# Emerging Biopharmaceuticals from *Pimpinella* Genus

**DOI:** 10.3390/molecules28041571

**Published:** 2023-02-06

**Authors:** Jiajia Wu, Zhen Cao, Syed Shams ul Hassan, Haozhen Zhang, Muhammad Ishaq, Xu Yu, Shikai Yan, Xue Xiao, Hui-Zi Jin

**Affiliations:** 1Shanghai Key Laboratory for Molecular Engineering of Chiral Drugs, School of Pharmacy, Shanghai Jiao Tong University, Shanghai 200240, China; 2Department of Natural Product Chemistry, School of Pharmacy, Shanghai Jiao Tong University, Shanghai 200240, China; 3Shanghai Institute of Pharmaceutical Industry Co., Ltd., China State Institute of Pharmaceutical Industry, Shanghai 201203, China; 4Institute of Chinese Medicinal Sciences, Guangdong Pharmaceutical University, Guangzhou 510006, China

**Keywords:** *Pimpinella*, natural products, phytochemistry, terpenoids, inflammation, medicine

## Abstract

Evolved over eons to encode biological assays, plants-derived natural products are still the first dawn of drugs. Most researchers have focused on natural compounds derived from commonly used *Pimpinella* species, such as *P. anisum*, *P. thellungiana*, *P. saxifrage*, and *P. brachycarpa*, to investigate their antioxidant, antibacterial, and anti-inflammatory properties. Ethnopharmacological studies demonstrated that the genus *Pimpinella* has the homology characteristics of medicine and food and mainly in the therapy of gastrointestinal dysfunction, respiratory diseases, deworming, and diuresis. The natural product investigation of *Pimpinella spp*. revealed numerous natural products containing phenylpropanoids, terpenoids, flavonoids, coumarins, sterols, and organic acids. These natural products have the potential to provide future drugs against crucial diseases, such as cancer, hypertension, microbial and insectile infections, and severe inflammations. It is an upcoming field of research to probe a novel and pharmaceutically clinical value on compounds from the genus *Pimpinella*. In this review, we attempt to summarize the present knowledge on the traditional applications, phytochemistry, and pharmacology of more than twenty-five species of the genus *Pimpinella*.

## 1. Introduction

Secondary metabolites from nature, predominantly plant, are still elected as a first preference for drug discovery and serve as a hotpot because of their promising novel scaffolds against chronic diseases. Once the only thirst to cure diseases, elixirs and traditional medications allow for the more proficient approach to drug discovery. Plant-derived natural products were once the backbone of the pharmaceutical armamentarium, but the ready corresponding access to synthetic agents has discouraged the interest in maintaining a discovery paradigm from plants.

Recently, drug discovery from plants has sparked in many researchers and they have driven back their path toward terrestrial plants. *Pimpinella* is a species richness genus in the Umbelaceae family with unique morphological characteristics of monofoliate or compound leaves, three-out or one- to two-fold pinnate division. The flowers possessed characteristic umbels white or purplish-red, and ovoid and long ovoid was the most common fruit shapes [1]. The morphological characteristics of several major *Pimpinella* plants distributed in China are shown in Figure 1. By reviewing the literature, we discorvered about 150 species of *Pimpinella* were widely distributed in the area of Asia, Europe, and Africa, while only a few species were discovered in north and west of North America [2]. China, Turkey, and Iran were the three most abundant distribution centers of different species [3,4]. Approximately 39 species have been recorded in the Chinese Flora, most of which were perennial herbs, and a few were biennial or annual herbs [5]. A total of 26 species were native to Turkey and 23 species were present in Iran. Among them, *P. anisum* is the representative folk medicine growing in Asia, Europe, Iran, and the United States with the most research reports, the highest application value, and the widest distribution range.

According to the theoretical knowledge of TCM (traditional Chinese medicines) and the recordation of Chinese Pharmacopoeia in the 2010 edition, the Chinese herbal medicine Yanghongshan (*P. thellungiana*) has the effects of warming the middle and dispersing cold, invigorating the spleen and replenishing qi, nourishing the mind and transcending the mind, relieving cough and removing phlegm, and clinically used to treat Keshan disease, palpitations, shortness of breath, and cough [1]. Phytochemistry investigation revealed that the genus *Pimpinella* principally contained compounds of terpenoids, flavonoids, coumarins, sterols, and fatty acids [6]. Pharmacological research has revealed a variety of biopharmacological activities of the extracts and compounds from the genus *Pimpinella*, such as antioxidant [7], antibacterial [8], anti-inflammatory [9], antitumor [10], and hypoglycemic activity [11]. However, the in-depth research on the traditional medicinal use of extracts from *Pimpinella* is insufficient presently, and the research on the chemical components and pharmacological effects is only focused on several species. Hence, more theoretical support for the clinical application and toxicity of *Pimpinella* is necessary.

Systematic retrieval of relevant literature was consulted on the following electronic databases: the Web of Science database (http://apps.webofknowledge.com/ (accessed on 8 June 2022)), the PubMed Database (https://pubmed.ncbi.nlm.nih.gov, (accessed on 8 June 2022)), Chinese National Knowledge Infrastructure (CNKI) (http://www.cnki.net, (accessed on 8 June 2022)), Wanfang Data (http://www.wanfangdata.com.cn/, accessed on 8 June 2022), and Google Scholar, and information only written in Chinese and English were considered. The keywords included “*Pimpinella*”, “Hui-qin” (in Chinese), “anise”, “constituents”, “separation”, and “pharmacology”. Additional data were collected from the relevant surveys of PhD. and MSc. research in China by the CNKI database, monographs on folk medicine, and Chinese Pharmacopoeia (2020). In this review, we systematically and comprehensively consulted and summarized over 100 references on the folk-medicinal application, phytochemical constituents, and pharmacological activities of the genus *Pimpinella*, providing a theoretical basis for promoting the application of *Pimpinella* plants in medicine, food, and other fields.

## 2. Folk-Medicine Application

The common ethnic uses of *Pimpinella* around the world are summarized as shown in Table 1, and it can be concluded that *Pimpinella* plants have the homology characteristics of medicine and food, wide varieties, and extensive traditional activities. In Asia, China was the country with the longest history and abundant resources in herbal remedy. *P. diversifolia* was used for the treatment of cold, indigestion, and diarrhea; *P. candollean* was eaten locally by Hmong as wild vegetables and used for resistance to stomach pain, bone pain, and rheumatism; *P. thellungiana* showed a remarkable anticoagulant effect [12]. Additionally, Koreans were keen to make *P. brachycarpa* delicious kimchi and were used medicinally for gastrointestinal dysfunction, asthma, and cough [13,14]. The seed of *P. monoica* native to India was used to fight stomachaches [15]. In the Middle East, species diversification of *Pimpinella* could be observed in Turkey. The recorded endemic species, *P. cappadocica* [16], *P. rhodantha* [17], *P. peregrine* [18], and *P. khorasanica* [19] were applied in the therapy of deworming, digestion, sedation, expectoration, and increasing lactation. In Iranian folk studies, *P. anisum* seeds treated epilepsy since ancient times [20], while residents in Egypt and Lebanon attempted to use it to treat digestive and respiratory ailments [21,22]. Notably, *Pimpinella*’s medicinal use is less prevalent in Europe and America, and the British and Brazilians usually used *P. anisum* as insect repellent, urinary disinfectant, and a deobstruent [23,24]. In Spain, France, and Italy, *P. anisum* was added to cooking, distilled alcoholic spirits, and confectionery industries as botanical spices [25,26].

## 3. Phytochemistry

Recent investigations on chemical constituents of the genus *Pimpinella* identified 343 compounds, principally containing phenylpropanoids, terpenoids, flavonoids, coumarins, sterols, and organic acids. More than 80% of compounds were identified after 2000 (Figure 2), among which phenylpropanoids, terpenoids, and flavonoids are the essential active components with the functions of an antioxidant, anti-inflammatory, and antitumor. Particularly, a unique and infrequent phenylpropanoid was found in the genus *Pimpinella*, named pseudo isoeugenol.

### 3.1. Phenylpropanoids

Phenylpropanoids are the main components occupying the dominant activity position in the volatile oils or extract of the *Pimpinella* genus, and a battery of studies have revealed anti-inflammatory, antibacterial, and antioxidant contributions [27]. Phenylpropanoids area class of natural compounds concatenated bya benzene ring and three straight-chain carbons (C_6_-C_3_ units) with diverse activities. According to the skeletal characteristics of the phenylpropanoids in *Pimpinella*, they could be divided into three groups: pseudoisoeugenol, pseudoisoeugenol derivatives, and simple phenylpropanoids. Pseudoisoeugenol, as a representative chemical marker of *Pimpinella*, possessed a peculiar skeleton of 1-hydroxy-2-propyl-4-methoxybenzene, which has been found exclusively in the genus *Pimpinella* so far [28]. Since the first pseudoisoeugenol was discovered from the *P. anisum* by G. T. Carter et al. [29] in 1977, scientists have successively obtained 12 pseudoisoeugenol (**1**–**12**) from the Chinese herb *P. thellungiana*, and llungianin A (**1**) and llungianin B (**2**), which presented significant antihypertensive activity [30,31,32,33,34,35,36,37].

Pseudoisoeugenol derivatives owned the same basic skeleton of a 1,2, 4-trisubstituted benzene ring as pseudoisoeugenol. The different was that the position of different-types substituent changed, and the phenolic hydroxyl group was combined with the acyl group to form an ester group frequently. Scientists have demonstrated the presence of 19 pseudoisoeugenol derivatives (**13**–**31**) in the genus *Pimpinella*, most of which were identified by GC-MS from the volatile oil [36,37,38,39,40,41,42,43,44,45].

In addition to the pseudoisoeugenol and its derivatives, the scientists have isolated 20 other types of phenylpropanoids from the genus *Pimpinella*. Two new phenylpropanoids (**32**–**33**) with antioxidant peculiarity were extracted by coupling with vacuum liquid chromatography and preparative thin-layer chromatography from *P. aurea* [45]. Additionally, eight simple phenylpropanoids (**34**–**46**) were sought during GC-MS analysis of essential oils from *P. anisum*, *P. corymbosa*, *P. peregrine*, *P. puberula*, *P. anisetum,* and *P. flabellifolia* gathered in Turkey [36,45,46,47]. Moreover, anisketone (**47**), methyl-*O*-coumarate (**48**), 1-(2-hydroxy-4-methoxyphenyl)-propan1-one (**49**), 4-methoxycinnamaldehyde (**50**) and dillapiole (**51**) were obtained from *P. saxifraga* and possessed antioxidant activity and DNA protection potential [46]. The phytochemical profile of *P. serbica*, endemic to West Balkans, was dominated by phenylpropanoids, dillapiole (35.1%) (**51**), and nothoapiole (9.5%) (**52**) [47] (Figure 3, Table 2).

### 3.2. Terpenoids

Terpenoids are a kind of active ingredient with diverse skeletons with manifold bioactivity and extensive distribution. The formula complies with the (C_5_H_8_)_n_ rule by polymerization of isoprene units in different linking ways. The long-term research by phytochemistshave certified that the volatile oil containing a large number of terpenoids has pervasively existed in *Pimpinella*, which released crucial activities of antibacterial, anti-inflammatory, and antidepressant activities. Notably, besides unique phenylpropanoids, the considerable quantity of specific C-12 sesquiterpene is another phytochemical marker distinguishing *Pimpinella* from other genera, such as geijerene- (**107**) and azulene- (**109**) type terpenes.

The volatile oil of *P. anisum* seed has attracted attention for its extensive biological activities such as antitumor, anti-inflammatory, and insecticidal agents. Numerous studies have characterized the ingredients in *P. anisum*, from which 46 monoterpenoids and sesquiterpenoids, including the principal ingredient linalool (**101**), were identified [6,50,51,52,53]. Many researchers have analyzed the essential oil extracted from the foreign-sourced *Pimpinella* genus. A. V-Negueruela et al. [44] disposed of the aboveground parts of *P. junoniae* in Spain to obtain the oil, and *α*-zingiberene (20.6%) and *α*-pinene (17.9%) were the most abundant among 26 volatile constituents. N. Tabanca et al. [36,45] evaluated essential oils extracted from roots, stems, leaves, and fruits of four *Pimpinella* species (*P. aurea*, *P. corymbosa*, *P. peregrine* and *P. puberula*) on GC-MS, and a total of 95 terpenoids were identified. Meanwhile, further data comparison discovered that the main components of each plant in different parts possess differentiation, while only the oil from the root had a higher similarity, containing large quantities of epoxy pseudoisoeugenyl-2-methyl butyrate (26.8–42.8%). A series of *Pimpinella* plant’s in vitro activity exploration, including the Turkish medicines *P. anisetum* [48], *P. flabellifolia* [48], and *P. enguezekensis* [51], the Iranian medicinal plants *P. affinis* [52] and *P. khorasanica* [19], the Indian herb *P. monoica* [15], and the Tunisian wild vegetable *P. saxifrage* [46], exhibited good antioxidant and antibacterial capacities, and GC-MS reports revealed terpenoids were dominant for their therapeutic effect. Recently, researchers also conducted profound studies on the oil of domestic *Pimpinella* plants. X. W. Xu [53] and E. M. Suleimen et al. [54] extracted the oil of *P. diversifolia* and *P. thellungiana* and identified 19 and 16 terpenoids, respectively.

In addition to terpenoids identified from volatile oils, S. Y. Lee’s long-term research on chemical constituents of *P. brachycarpa* [13], two new sesquiterpenes (**152**–**153**) and ten known terpenes (**88**–**91**, **128**, **154**–**158**) were isolated from the methanol extract of aerial parts. Ozbek et al. [16,17] obtained a new trinorguaian-type sesquiterpenoid (**114**) and a newly discovered triterpenoid glycoside (**219**) from *P. cappadocica* and *P. rhodantha*, respectively. Six triterpenoids (**213**–**218**) were isolated from *P. anisum* aqueous extract [55,56]. Pimpinelol (**205**) [10], a novel irregular sesquiterpene lactone from *P. haussknechtii* could significantly restrain the vitality of human breast cancer cells by inducing endoplasmic reticulum stress (Figure 4, Table 3).

### 3.3. Flavonoids and Their Glycosides

Flavonoids are the focused topic of natural product excavation currently, and the majority of flavonoids in *Pimpinella* plants exhibited satisfactory antioxidant power in accordance with previous studies. Moreover, flavonoids showed more diversified bioactivities related to different functional groups, including phenolic hydroxyl, glycoside, and isopentyl. So far, more than 36 flavonoids have been isolated from this genus.

A series of investigations focused on the chemical compositions of TCM’s *P. thellungiana* in the past 40 years identified eight flavonoid glycosides (**220**–**227**) [60,61,62,63,64]. X. Chang [49] and J. Lu [12] systematically conducted the componential survey on *P. candolleana* and *P. brachycarpa,* respectively, and isovitexin (**228**), quercetin-3-*O*-rhamnoside (**229**), luteolin (**244**), and 1-hydroxy-2, 3, 5-trimethoxy xathone (**255**) were isolated. H. Ozbek et al. [16,17] obtained a series of flavonoid glycosides (**230**–**239**, **251**–**252**) with preferable antioxidant activity from Turkish *P. cappadocica* and *P. rhodantha* during in 2015–2016, including one novel *β*-hydroxy dihydro chalcone glycoside structure, ziganin (**251**) and three first-discovered acylated-flavonol glycosides, erzurumin (**236**), ilicanin (**237**), and quercetin-3′-methylether-3-*O*-*α*-L-(2″,3″-di-*O*-*trans*-coumaroyl)rhamnopyranoside (**238**), which enriched the flavonoid library of the *Pimpinella* genus. In 2020, the phytochemical profile of another traditional medicinal plant in Turkey, *P. anthriscoides*, was characterized by G. Zengin, et al. and four unique flavonoids, luteolin-7-*O*-glucoside (**240**), chrysoeriol-7-*O*-glucoside (**241**), diosmetin-7-*O*-rutinoside (**242**), and chrysoeriol (**243**), were identified [65]. A 2021 reporton analysis of *P. anisum* seed revealed the presence of eight flavonoids, including myricetin (**245**), quercetin (**246**), apigenin (**247**), kaempferol (**248**), chrysin (**249**), galangin (**250**), naringenin (**253**), and epigenin (**254**) [66] (Figure 5, Table 4).

### 3.4. Coumarins

Coumarins are widely distributed in Umbelliferae, Rutaceae, Asteraceae, Leguminosae, and Solanaceae, and 25 coumarins have been found in the *Pimpinella* genus in phytochemical relevant studies. The domestic scholar B. L. Qiao et al. [67], separated five components from the ethyl acetate extract of *P. thellungiana*, which were identified as coumarins after structural identification, bergapten (**256**), marmesin (**257**), scoparone (**258**), scopoletin (**259**), and isofraxidin (**260**). Subsequently, P. Pradhan and his team weresurprised to find a novel skeleton of natural furanthochromone dimers or oligomers, including visnagin (**261**), pimolin (**262**), visnagintrimer (**263**), visnagin tetramer (**264**), visnagin pentamer (**265**), and khellin (**266**), from chloroform extract of seeds in *P. monoica* [68,69]. Additionally, the phytochemical profile of *P. anthriscoides* was characterized and aegelinol (**267**), psoralen (**268**), imperatorin (**269**), is oimperator in (**270**), 3-(1,1-dimethylallyl)herniarin (**271**), peucedanin (**272**), and xanthyletin (**273**) were identified [65]. Some investigations [70,71,72,73] found that abundant linear coumarins (**274**–**280**) existed in the root and seed extract of *P. anisum*, among which isopimpinellin (**274**) and methoxsalen (**275**) were associated with the inhibitory activity of the cytochrome P450 1A2 isozyme in healthy adults. In addition, umbelliprenin (**276**) was proven to be an original skin-whitening agent (Figure 6, Table 5).

### 3.5. Sterols

Phytosterols are nutritional compounds from *Pimpinella* plants equipped with capabilities of cholesterol-reducing, blood pressure-lowering, and anti-inflammatory properties. Surprisingly, the variety and activity of sterols in *P. anisum* are research worthy. R. K. Saini, et al. acquired phytosterol profiling of *P. anisum* seeds by GC-MS measure [74]. It was surveyed that the total sterol content in seeds oil was 551.9 mg/100 g, and five sterols were identified, including the dominant ingredient α-spinasterol (**282**) (109 mg/100 g oil; 19.9% of the total sterols), campesterol (**281**), stigmasta-5,7,22-trien-3-ol (**283**), Δ7-avenasterol (**284**), and Δ5-avenasterol(**285**). Consistent with Saini’s results, S. Balbino, I. B. Rebey and M. Kozlowska, et al. [58,75,76] isolated affluent phytosterol compounds, i.e., Δ7-stigmastenol (**286**), Δ5, 23-stigmastadienol (**287**), Δ7-campesterol (**288**), sitostanol (**289**), cycloartenol (**290**), and 24-methylenecycloartenol (**291**) from *P. anisum*, recommending it as a natural source of salutary phytosterols. Furthermore, *b*-sitosterol (**292**) and *g*-sitosterol (**293**) were isolated from *P. thellungiana* and exhibited hypotensive activity [41]. Stigmasta-5, 22-dien-3-olacetate (**294**), daucosterol (**295**), and stigmasterol (**296**) were uncovered in *P. candolleana*, which displayed effective antioxidant and *α*-glucosidase inhibitory [49,75]. 24ζ-Methyl-5R-lanosta-25-one (**297**) and pregnenolone (**298**) provided an antioxidant property derived from *P. brachycarpa* [76] (Figure 7, Table 6).

### 3.6. Organic Acids

Organic acids are widely distributed in leaves, roots, and fruits, and as aromatic plants, organic acid is a crucial element of volatile oil in the *Pimpinella* genus. Its structural types included aliphatic polycarboxylic acid, aromatic benzoic, and caffeic acid with anti-inflammatory and antioxidant biological properties. Six reports [34,42,63,79,80,81] were performed by chemical separation, HPLC fingerprint characterization, and UHPLC-Q-Orbitrap HRMS rapid identification, and a total of 17 organic acids (**299**–**315**) were identified from *P. thellungiana* with abundant quinic acid derivatives. In other reports, five new quinic acid derivatives, 1-*O*-*trans*-caffeoyl-5-*O*-*trans*-coumaroylquinicacid (**316**), 1-*O*-*trans*-caffeoyl-5-*O*-7,8-dihydro-7*α*-methoxy caffeoy lquinic acid (**317**), 1-*O*-7,8-dihydro-7*α*-methoxycaffeoyl-5-*O-trans-*caffeoylquinic acid (**318**), 1,5-di-*O*-*cis*-coumaroylquinic acid (**319**), and 1,5-*O*-*trans*-dicaffeoylquinic acid (**320**), together with 10 known quinic acid derivatives (**306**–**310**, **313**, and **321**–**324**) with anti-neuroinflammatory activity, were isolated from the methanol extract of *P. brachycarpa* [82]. A. Topcagic, et al. identified 12 phenolic acids (**325**–**336**) from *P. anisum* during the analysis of volatile oil [66]. Moreover, 3-phenyllactic acid (**337**) and citric acid (**338**)were obtained from *P. anthriscoides* and had a proven antioxidant and inhibiting *α*-amylase, *α*-glucosidase, AChE, and BChE effect [65]. Long-chain fatty acids, tetradecanoic acid (**339**), linoleic acid (**340**), and stearic acid (**341**) were isolated from the volatile oil of *P. diversifolia* leaves [53], while dodecanoic acid (**342**) and pentadecanoic acid (**343**) were identified from *P. aurea* oil [36] (Figure 8, Table 7).

## 4. Pharmacology

Since the beginning of this century, due to the extensive use of the genus *Pimpinella* in traditional medicine, numerous scientific studies have demonstrated several ethnopharmacological properties from its extracts or compounds, including antibacterial, anti-inflammatory, insecticidal, antioxidant, and inhibitory enzyme activities [83]. In addition, some novel pharmacological activities such as antitumor, antidepressant, blood pressure lowering, hypoglycemic, and liver protection have been gradually exploited recently. In our review, the effect of the *Pimpinella* species during the recent 8 years (2015–2022) was summarized, and specific pharmacological studies were discussed in the following paragraphs, as presumptively presented in Table 8 (Figure 9).

### 4.1. Antioxidant Activity

Plant-derived compounds have promising antioxidant activities (1–2). Within the past eight years, twenty studies have revealed the antioxidant properties in *Pimpinella* species, concentrating on *P. anisum* (60% of all studies). Seeds (70%) and aboveground parts (25%) were considered to be admirable candidates as antioxidants, and aromatics and flavonoids were identified as the dominant components. Experiments were divided into two categories: in vivo level and in vitro level. In vitro activity screening was a rapid and efficient antioxidant assay, with the precedence of animal studies constituting 90% of all tests.

Since 2015, only two in vivo tests were reported relating to the antioxidant activity of *P. anisum*. Favism is a metabolic disease of acute hemolytic anemia induced by bean consumption. In 2016, Kori, et al. demonstrated that pretreatment with *P. anisum* oil could block the oxidative stress effect of the causative agent to achieve a favism-protective effect by arresting the hydrolysis-of vicine ran convict to their aglylate free radical compounds (divicine and isouramil), and this effect related to anethole [7]. Ashtiyani’s et al, study was aimed at exploring the alleviating effect of a *P. anisum* ethanol extract on gentamicin (GN)-induced Wistar rat model of nephrotoxicity by interfering with oxidative stress [84]. bThe results showed that *P. anisum* reversed the GN-induced increase in levels of plasma creatinine, BUN, MDA, and excretion of sodium and potassium and improved FRAP and GN-induced tubule damage.

On the other hand, the in vitro antioxidant performance of *P. anisum* was evaluated by utilizing different radical scavenging activities, such as DPPH and ABTS, reducing capacity assay (FRAP and PMCA), and *β*-carotene/linoleic acid determination. Many types of research showed satisfactory antioxidant properties of ethanol extract [84], aqueous extract [56], *n*-hexane extract [11], and volatile oil [50,85,86] of *P. anisum* seeds by various tests. As expected, further data comparison indicated that the DPPH clearance rate of oil exceeded 77% at the optimal concentration, superior to other types of extracts, and is recommended as a natural antioxidant. Furthermore, another analysis of oxidative–correlative components revealed that *P. anisum* oil possessed a positive correlation with the total amount of phenols and polysaccharides [9] and a negative correlation with the total amount of sterols [78].

Meanwhile, many studies clarified the antioxidant effect of different *Pimpinella* species abroad, providing a logical basis for the rational choice of the *Pimpinella* plant. Ozbek et al. also proved the superior antioxidant activities of *P. cappadocica* [16] and *P. rhodantha* [17], which were consistent with the flavonoid glycosides content, while the antioxidant capacities of *P. enguezekensi* [51] and *P. anthriscoides* [65], newly discovered species in eastern Anatolia, that were attributed to high *trans*-anethole concentration. The antioxidant characterizations of ethyl acetate extracts from Indonesian *P. alpine* [88] and Iranian *P. affinis* [89] were conducted by in vitro screening with IC_50_ values of 53.07 and 74.90 µg/mL, respectively. A study in 2019 [46] discovered that 3% of *P. saxifraga* oil exhibited significant antioxidant activity and DNA protection potential, correlating with the proportion of phenolic compounds [90], which indicated it could be used as a new natural antioxidant candidate added to the sodium alginate coating in the preservation of cheese.

### 4.2. Antibacterial Activity

Bacterial infection is the main cause of morbidity and mortality throughout the world. Since antiquity, scientists have been interested in its bacteriostatic potential due to the characteristic volatile compositions in the *Pimpinella* species. In Table 8, most data concerning *P. anisum* oil presented that phenyl propanes, especially anethole and its isomers, were the predominant components accounting for 98% of its content [149]. Various test procedures were conducted, such as disk diffusion, agar diffusion, minimum inhibitory concentration (MIC), and minimum bactericidal concentration (MBC) using in vitro conditions to explore the antagonistic activity of microorganisms of extracts from different species.

Since 2015, 19 reports have multidimensionally characterized the antimicrobial activity of *P. anisum*, accounting for 79%. The essential oil from *P. anisum* has been triumphantly developed as a target preparation, and with advances in biological materials, the combination of PLA film materials, nano emulsions, and gel materials with oil has been affirmed as a new dosage form, which could greatly improve its antibacterial ability. In terms of antibacterial experiments, many studies demonstrated that oil and polysaccharide from *P. anisum* seeds and fruits exhibit antibacterial activity against a battery of gram-negative and gram-positive bacterium (Table 8) [8,9,91,92,150]. Noteworthily, fire blight was a devastating disease of commercial crops of *Rosaceae*, ascribing to the highly infectious bacteria *Erwiniaamylovora*, and Akhlaghi et al. found that *P. anisum* oil showed above-average antibiotic ability with a MIC of 31.25 μg/mL [93].

In antifungal experiments, *P. anisum* oil-hydrogel formulation was successfully prepared by the freeze-drying method, which was suitable for vaginal delivery systems and showed restraining activity against *Candida albicans*, *C. glabrata*, and *C. parapsilosis* [94]. Currently, aromatic plants have attracted interest for scientists as sources of natural antimicrobials due to the increased resistance of pathogenic fungi. Khosravi et al., confirmed *P. anisum* oil was sensitive to *Fusarium solani* emerging from patients with onychomycosis with a MIC ranging from 50 to 490 μg/mL [95].

In another dermatophyte infections study [96], combined treatment with terbinafine and *P. anisum* oil showed that oil enhanced the activity of terbinafine against *Trichophytonrubrum* and *T. mentagrophytes* with a 4-fold reduction in the MIC. The combination therapy had a synergistic effect on reducing the concentration of antifungal drugs and the appearance of resistant strains than monotherapy. A. J. Obaid et reported that *P. anisum* oil down-regulated the keratinase gene expression of *T. rubrum* by 0.079 compared with control (1.00), conducing to target determination during drug development [97].

In recent years, with the continuous improvement of consumer requirements for food safety, the application of *P. anisum* oil as a natural antibacterial agent has been greatly promoted in the food domain. Many microbiology experiments [98,99] demonstrated that *P. anisum* oil exerted an inhibitory effect on the growth of the food-born germ *Clostridium perfringens* and several mycete by controlling of mycelium growth and spore germination [100]. Khoury et al. further integrated with qRT-PCR to reveal the modulation of 5 µL/mL *P. anisum* oil on the ochratoxin A production during grape brewing by down-regulating the expression of *Aspergillus carbonarius* biosynthesis-related genes (*acOTApks*, *acOTAnrps*, *acpks* gene) and growth-regulating genes (*laeA* and *vea* gene) [101]. Noori et al., research in 2021 showed a concentration-dependent inhibitory effect on *Listeria monocytogenes* and *Vibrio parahaemolyticus* by adding *P. anisum* oil to a novel polylactic acid (PLA)-based composite film for food packaging [102]. Ultrasound-assisted *P. anisum* oil-based nanoemulsion prevented microbial contamination induced by 5 bacteria and 14 food-contaminating fungi compared with pure extract and is recommended as a green food antiseptic [87,105,106,107].

In addition to *P. anisum*, many studies exhibited the antimicrobial potentials of different *Pimpinella* species from around the world, including, *P. alpine* [88], *P. saxifrage* [46], *P. enguezekensis* [51], *P. affinis* [52], and *P. anthriscoides* [65], which showed the moderate bacteriostatic effect against a battery of microorganisms, suggesting development as an alternative for *P. anisum*.

### 4.3. Anti-Inflammatory Activity

The cause of body inflammation is either infection or physical/chemical damage. In that case, blood starts oozing out into tissues from blood vessels (5–6). *P. anisum* has been approved by the Committee of Herbal Products of the European Medicines Agency (EMA) for a therapeutic schedule of mild indigestion and an expectorant for coughs due to its traditional effects on respiratory disorders. As the literature ascertained, the genus *Pimpinella* exerted an anti-inflammatory effect by regulating the expression of proinflammatory cytokines (IL-1, IL-8, and TNF-α), and anethole (**40**) from the volatile oil was the prime ingredient [96]. However, there was little research on its anti-inflammatory mechanism in respiratory tissues. T. P. Domiciano et al. previously provided preclinical evidence that anethole (**40**) inhibited the production or release of PGE_2_ and NO in acute inflammation in animals [106]. R. Iannarelli et al. [107] further revealed that *P. anisum* oil acted as a remarkable anti-inflammatory by reducing the expression of IL-1 and IL-8 in LPS-induced tracheal epithelial HBEpC and HTEpC lines and promoting the secretion of Muc5ac. Another study on the respiratory system examined the effects of anethole (**40**) on the inflammatory status of lung and liver cells after exposure to airborne pollution of particulate matter (PM). In PM_2.5_-induced BEAS-2B and HepG2 cells, anethole (**40**) reduced the levels of IL-6 and IL-8 by 96% and 87%, respectively, demonstrating it is a natural therapeutic agent to counteract PM-induced inflammation [108]. Recently, based on analysis of ovalbumin (OVA)-induced allergic rhinitis (AR) model mice, C. S. Liao’s team found that the anti-inflammatory response of BLAB tea containing *P. anisum* was relevant to the suppression activity on the accumulation of inflammatory cells and the release of Th2 and histamine in the nasal mucosa, NALF, and serum, and induction of the production of Th1 and Treg [109]. Another *P. anisum* study [9] indicated that polysaccharide extract mediated anti-inflammatory effects by improving edema and reducing MDA and SOD levels of oxidative stress indexes in muscle in carrageenan-induced foot swelling in mice.

### 4.4. Anti-Tumor Activity

It is important to note that the antitumor activities of genus *Pimpinella* have been verified at the cellular level and in animal studies, while few studies report on clinical applications. Terpenoids from *P. anisum* seed are the dominant antitumor compounds. A 2015 study showed that treating HepG2 cells with *P. anisum* oil for 24 h caused concentration-dependent and significant cytotoxicity [110]. Nowadays, silver nanoparticles (AgNPs) provide a new pathway for the utilization of natural products and the importance of drug release. Alsalhi et al., designed a green synthetic route in 2016 to prepare AgNPs containing a *P. anisum* aqueous extract, which exerted obvious antitumor effects on human neonatal skin stromal cells and colon cancer cells [111]. S. Devanesan et al. conducted an in-depth study on the pharmacological mechanism of AgNPs in the colorectal cancer cell (CRC) [112]. Interestingly, synthetic AgNP could selectively destroy CRC via the inhibition of proliferation, arresting the cell cycle at the G2/M phase, and inducing apoptosis, indicating that composite nanomedicines may pioneer new approaches for prospective anticancer therapy. A. Mahmoud et al. reported a novel sesquiterpene lactone pimpinelol (**205**) from *P. haussknechtii* and demonstrated reduction viability against human breast cancer cell line (MCF-7 cells, IC_50_: 1.06 μM) by inducing protein aggregation and endoplasmic reticulum stress at the cytokine levels [10].

### 4.5. Hypoglycemic Activity

Diabetes is a lifelong metabolic disease characterized by hyperglycemia, leading to a variety of deadly complications. Previous reports have confirmed that the ethanol extract of *P. brachycarpa* possesses the capacity for precaution of hyperglycemia and remission of oxidative stress in type II diabetic mice [14]. Since 2015, studies paid attention to *P. anisum* in controlling hyperglycemia and preventing diabetes complications. Preliminarily, M. Bonesi et al. evaluated the inhibitory activity of *P. anisum* seed on two key enzymes associated with type II diabetes, and it exhibited moderate inhibition against *α*-amylase and *α*-glycosidase with IC_50_ values of 692.6 ± 5.2 and 73.9 ± 2.2 μg/mL, respectively [11]. Secondly, a 2020 study using a streptozotocin (STZ)-induced diabetic rat model observed that β-cell structure was significantly improved, insulin immune response was enhanced, and pancreatic acinus and amylase levels were reduced in the *P. anisum*-treated group compared to diabetic-control. The authors attributed the beneficial effects of *P. anisum* extract to its hypoglycemic and antioxidant properties, as oxidative stress plays a critical role in the development and progression of diabetes. In this study, the *P. anisum*-treated group significantly reduced SOD and CAT and increased their levels of lipid peroxidation marker MDA, which plays a role in lowering blood glucose. In addition, in immunohistochemical experiments, it could be observed that compared with diabetic control groups, the caspase 3 immunoreaction (22.34 ± 1.27 vs. 52.96 ± 2.32) and beclin 1 immunoreaction (31.55 ± 1.05 vs. 46.85 ± 1.30) were significantly decreased in the *P. anisum*-treated group (*p* < 0.001). These results indicated that *P. anisum* could significantly down-regulate the autophagy regulation marker beclin 1 and apoptosis marker caspase 3 in the pancreas, also relating to its antioxidant properties. Finally, M. Hashemnia et al. explored the potential of *P. anisum* on skin ulceration complications induced by diabetes from a new perspective of wound healing [114]. *P. anisum* reversed oxidation changes of MDA and GSH in wound skin (*p* < 0.05) and significantly reduced the wound size and the number of inflammatory cells while enhancing the re-epithelialization rate, collagen content, and fibroblast reaction, promoting festering wound reparation in diabetic rats.

### 4.6. Hypotensive Activity

A report in 2017 demonstrated that the ethyl acetate and ethanol extract of *P. brachycarpa* has a significant antihypertensive function in hypertensive model rats [115]. Further invitro exploration revealed it exhibited a dose-dependent inhibitory effect on an angiotensin-converting enzyme in the range of 0.5–10 mg/mL, and 80% ethanol extract presented the highest inhibitory rate. However, the effective ingredients and mechanism of hepatoprotective activity need to be further clarified. Another study in 2019 confirmed that an aqueous extract of *P. anisum* seed had the beneficial effect of lowering arterial blood pressure in rats and further explored its mechanism by estimation of different models [116]. V.B.C. Pontes et al. successively eliminated the actions of diuresis, angiotensin receptor antagonism, and *β*-receptor blockade of *P. anisum*. Additionally, it proved to act as a calcium channel antagonist to act as a hypotensive agent by inhibiting Ca^2+^ influx.

### 4.7. Insecticidal Activity

Since the 20th century, the wide application of pesticides has led to the rapid development of agriculture and a booming increase in output. However, the increasing pests’ resistance and soaring pollution in the environment and food caused by synthetic pesticides have motivated researchers to explore natural botanicals as sources of new insecticides, such as *Pimpinella* essential oils.

A 2018 study [117] used *P. anisum* oil to explore the toxicity of agricultural pests and the safety of beneficial insects, and the results displayed noteworthy insecticidal effects against two pathogenic insects, *Culex. quinquefasciatus* (LC_50_ = 25.4 μL/L) and *Scaphoideus littoralis* (LD_50_ = 57.3 μg/L). Nevertheless, it was not toxic to beneficial insects in comparison with *α*-cypermethrin at the same lethal concentration. Similar inferences were drawn from another nine studies by contact and fumigation tests [118,119,120,121,122,123,124,125,126]. A.Hatege kimana et al. revealed another pathway in the eradication of the pest (*Acanthoscelidesobtectus*) by reducing fecundity (egg production) and fertility (egg hatch ability/progeny production) [127]. Ulteriorly, in vitro tests observed activity decline of AchE in two-spotted spider mites after *P. anisum* management, which was attributed to the high-content ingredients, such as *E*-anethole, isoeugenol, and *α*-pinene [86]. In addition, green insecticides with the cooperation of emerging eco-friendly substances and natural ingredients are perceived as a strategy. K. A. Draz et al. prepared *P. anisum* oil of nanoemulsions (2500 mg/L) to eliminate the emergence of *Sitophilus oryzae* and *Triboliumcastaneum* by 94.6% and 84.5%, respectively, which exceeded the values compared to that of pure oil; it had no adverse impact on the germination rate of wheat [128]. Concurrently, various attempts [129,130,131,132,133] have proved *P. anisum*-nanoformulation possessed considerable repellent and toxic activities against *Bactroceraoleae* and other crop pests.

In addition to the management of crop pests, *P. anisum* oil had an excellent performance on larval elimination and cutting-off transmission against pests spreading epidemic diseases. Numerous studies have provided convictive evidence of larval killing and oviposition deterrent activities of *P. anisum* on pestiferous pests, the vector of dengue, human African trypanosomiasis, and filariasis [48,136,137,138,139,140]. A. T. Showler et al. further demonstrated *p*-anisaldehyde was a botanical ingredient inhibiting the reproduction of pests [139,140]. To develop efficient mosquitocide, S. S-Gomez et al. encapsulated *P. anisum* oil in nanoparticles loaded with zein to overcome the defects of high degradability and low persistence and successfully applied it to mosquito larvicide [141]. Overall, *Pimpinella* oil not only combated insect vectors but also prevented crops and other organisms from toxic damage, representing a milestone in the commercial development of green-based insecticide formulations.

### 4.8. Enzymes Inhibitory Activity

O. H. Chan’s research declared that the ethanol extract of *P.brachy carp* regulated the enzymes CYP1A2, 2B6, and 3A4 by concerted inhibition, while it affected the enzymes CYP2C19 and 2D6 by competitive inhibition [142]. Furthermore, G. Zengin et al. evaluated the enzymatic inhibition of Turkish *P. anthriscoides* on tyrosinase, *α*-amylase, *α*-glucosidase, AChE, and BChE by invitro tests [65]. Since 2015, six studies focused on exploring the *P. anisum*-derived enzyme inhibitor. On one hand, scientists actively probed plant extracts, and a sol-gel GSTA1–1macroarray high-throughput screening tool was independently developed for celerity determination of the GST-inhibitory activity of *P. anisum* (IC_50_ = 3.40 ± 0.83 μg/mL) [143]. Gout was induced by excessive accumulation of uric acid due to xanthine oxidase (XO), which has the function of oxidizing hypoxanthine to xanthine and uric acid in an overactive state. L. Bou-Salah et al. revealed that *P. anisum* oil inhibited the activity of human-original XO (IC_50_ = 2.37 ± 0.23 μg/mL), discussing new tactics for gout treatment [144]. RALDHs were assigned to convert retinaldehyde to retinoic acid (RA), acting as the dominant mechanism in RA signaling pathways and relevant cancers. The current study indicated that ethanol extracts of *P. anisum* selectively and intensively inhibited RALDH3 expression, while it did not modulate RALDH1 and RALDH2, highlighting the selectivity of that in the regulation of RALDHs and the RA-governed metabolic process [145]. On the other hand, studies on *P. anisum* ingredients found that abundant bergapten (**256**), iso-pimpinellin (**274**), and methoxsalen (**275**) were inhibitors of CYP-1A2, which are involved in drug metabolism and carcinogenic bioactivity [70]. However, phenolic ingredients exerted remarkable inhibition against AChE and BChE with IC_50_ values of 0.07 and 0.34 μg/mL, respectively [56,66].

### 4.9. Antidepressant Activity

Depression and anxiety disorders are commonly believed to be stress-related mood disorders, invariably accompanied by various diseases and premature aging in severe cases. Many studies have manifested that the antidepressant effect of genus *Pimpinella* extracts is associated with neurotransmitters, genetic polymorphisms, endocrine system abnormalities, and cytokine levels [151]. A reversion of anxiety and depression and amelioration of memory formation in model mice by total extract of *P. anisum* [152] (100 mg/kg) and volatile oil of *P. peregrine* [18] could be observed according to researchers. However, precise elucidation of the mechanism was needed. Therefore, the in-depth study focusing on *P. anisum* oil by Koriem et al. [146] found that levels of 5-HT, DA, NE, GABA, and IL-10 were significantly reduced (*p* < 0.001) and the levels of inflammatory cytokines IL-1β, IL-6, TNF-α, and Ki-67 were significantly increased (*p* < 0.001) after oral administration of *P. anisum* oil compared with chronic mild stress (CMS) model rats, bringing the cerebral cortical and hippocampal levels close to normal. As is known, the inflammatory factors mainly occurred in allergy conditions. TNF-α represented an inflammatory factor in neurons where IL-1β produced inflammation through monocytes and macrophages; IL-6 and IL-10 had a vital role in the neuronal response to injury, while Ki-67 represented a nuclear protein, which was associated with cellular multiplication. These results confirmed the efficacy of *P. anisum* oil in the treatment of depression by inhibiting the inflammation of the cerebral cortex and hippocampus. It is worth noting that El-Shamy et al. concluded with conflicting results compared to Koriem et al., using the same animal models and experimental procedures as they attributed the depression-improving effect to its antioxidant activity [147]. The reason was decreased levels of GSH-Px, GST, GSH, and CAT, while increased levels of MDA and NO were observed in the cerebral cortex and hippocampus. Additionally, M-jahromi et al. [148] selected 120 patients with depression suffering from irritable bowel syndrome (IBS) to provide clinical evidence of the antidepressant effect of *P. anisum*. The *P. anisum*-treated group preferentially alleviated mild or moderate depressive symptoms in IBS patients compared to control and placebo groups making it a prospective and economical option for depressive patients.

### 4.10. Other Activities

In addition to the above pharmacological activities related to traditional usage, many novel pharmacological properties have been excavated from the genus *Pimpinella*. For example, the ethanol extracts of *P. anisum* combated uterine contractions by inhibiting *L*-type Ca^2+^ channels and blocking Ca^2+^ influx [153], and the polysaccharide extract accelerated wound healing [9]. Mosavata and his team implemented placebo-controlled trials to demonstrate that *P. anisum* ameliorated the distress of migraine [154] and premenstrual syndrome [155]. Moreover, umbelliprenin (**276**) in *P. anisum* has been proven to be a potential skin-whitening agent [71]. These data were anticipant of genus *Pimpinella* for drug exploitation in the treatment of various diseases.

## 5. Conclusions and Perspective

In this review, the traditional uses, chemical constituents, and modern pharmacological activities of the genus *Pimpinella* were summarized. Conclusively, genus *Pimpinella* principally contained phenyl propanoids, terpenoids, flavonoids, coumarins, sterols, and organic acids with a broad spectrum of biological activities, such as antioxidant, antibacterial, anti-inflammatory, antitumor, hypoglycemic, hypotensive, insecticidal, inhibitory enzyme, and antidepressant activities. Some *Pimpinella* cultivars could be applied as natural sources of edible vegetables, and essential oil was the important raw material for the production of green insecticides and condiments of alcoholic beverages.

This review is prepared to provide an overview of the knowledge of the last eight years (from 2015 to 2022) and to make suggestions for filling the gaps available in the literature for this genus. However, there were still some shortcomings during the overview of the genus *Pimpinella,* and suggestions were made for filling the gaps available in the literature for this genus. The mechanism, target, toxicity, and clinical application of the pharmacology needed to be further studied and discussed. Firstly, the species *Pimpinella* were abundant with similar appearance in China, and most were used as folk medicine. Detailed identification and quality standard were conducted only in *P. anisum* and *P. thellungiana*. Hence, it is urgent to establish a complete quality standard for *Pimpinella* plants to prevent the mixed-use phenomenon. Secondly, despite the increasing demand for pharmacological research on the genus *Pimpinella* recently, such as antioxidant, anti-inflammatory, antitumor, anti-depressant, and hypoglycemic effects, more attention should be paid to the relevant clinical research. The therapeutic properties recorded in medical books of various countries of all ages should be appreciated. For example, the traditional curative effect of *P. anisum* in the gastrointestinal tract and digestive function documented in many places has not been confirmed by uniting with modern scientific methods, which provides new directions for the future. Finally, *P. anisum*’s essential oil, aqueous, or organic solvent extracts are often applied for pharmacological investigation. To better clarify the pharmacological activity of *P. anisum*, the bioactivity-oriented separation method can be adapted to excavate the bioactive components and maximize utilization.

## Figures and Tables

**Figure 1 molecules-28-01571-f001:**
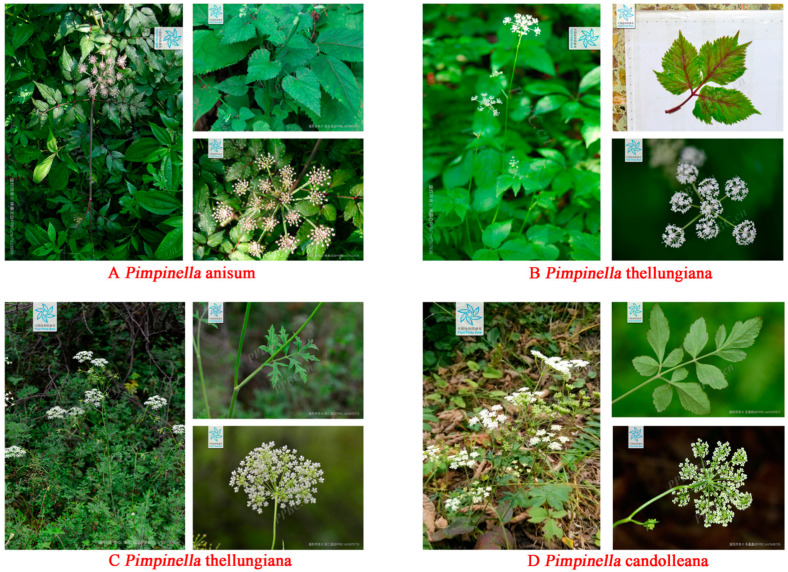
Morphological characteristics of several major Pimpinella species: (**A**), *Pimpinella anisum*; (**B**), *Pimpinella brachycarpa*; (**C**), *Pimpinella thellungiana*; (**D**), *Pimpinella candolleana* (Cited from http://ppbc.iplant.cn, (accessed on 8 June 2022)).

**Figure 2 molecules-28-01571-f002:**
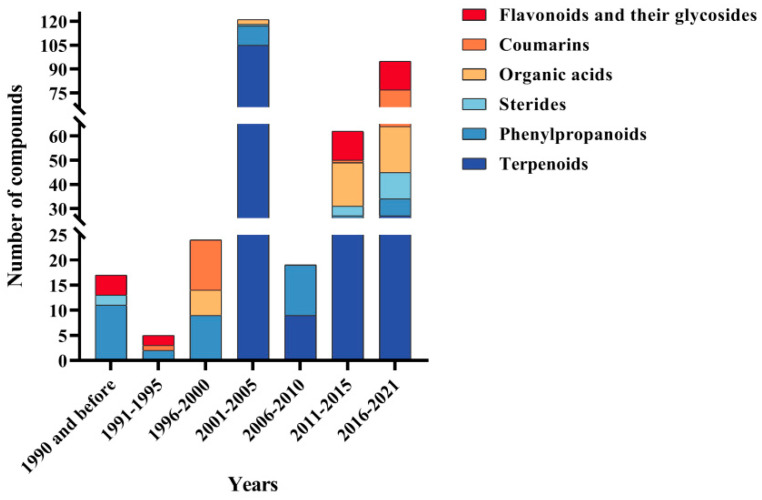
The number of different types of compounds identified from genus *Pimpinella* in different years.

**Figure 3 molecules-28-01571-f003:**
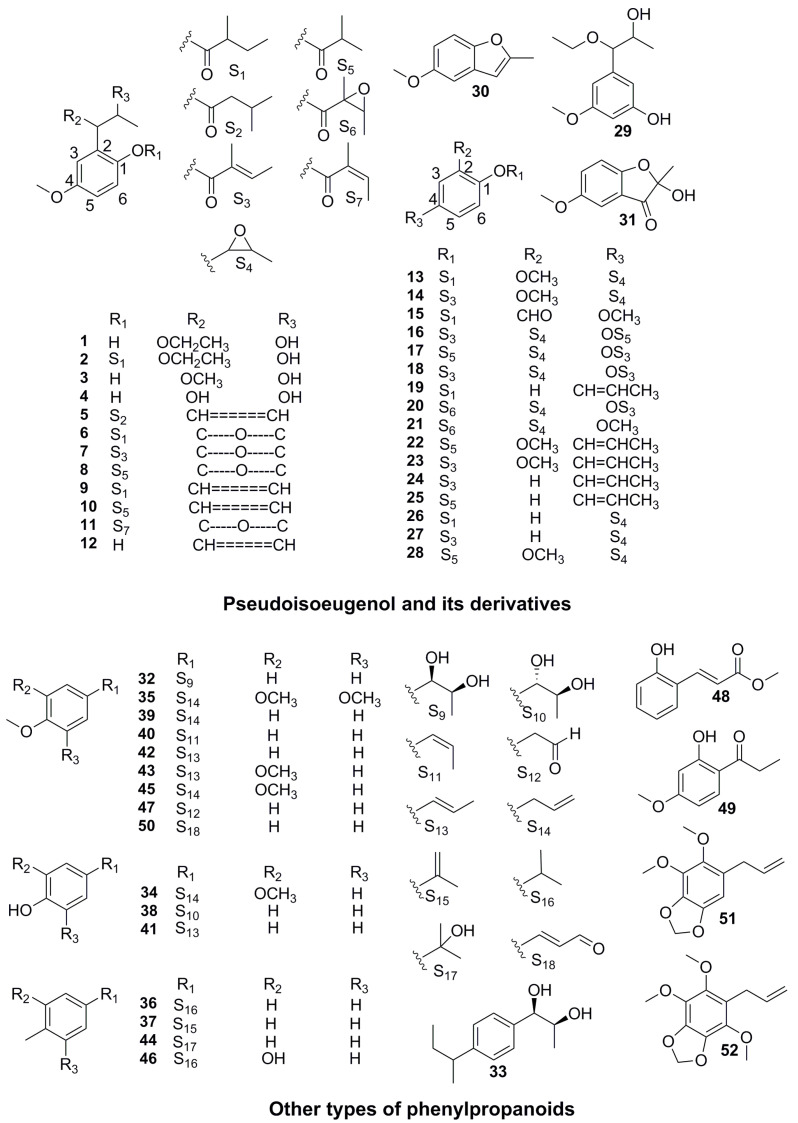
Chemical structures of phenylpropanoids.

**Figure 4 molecules-28-01571-f004:**
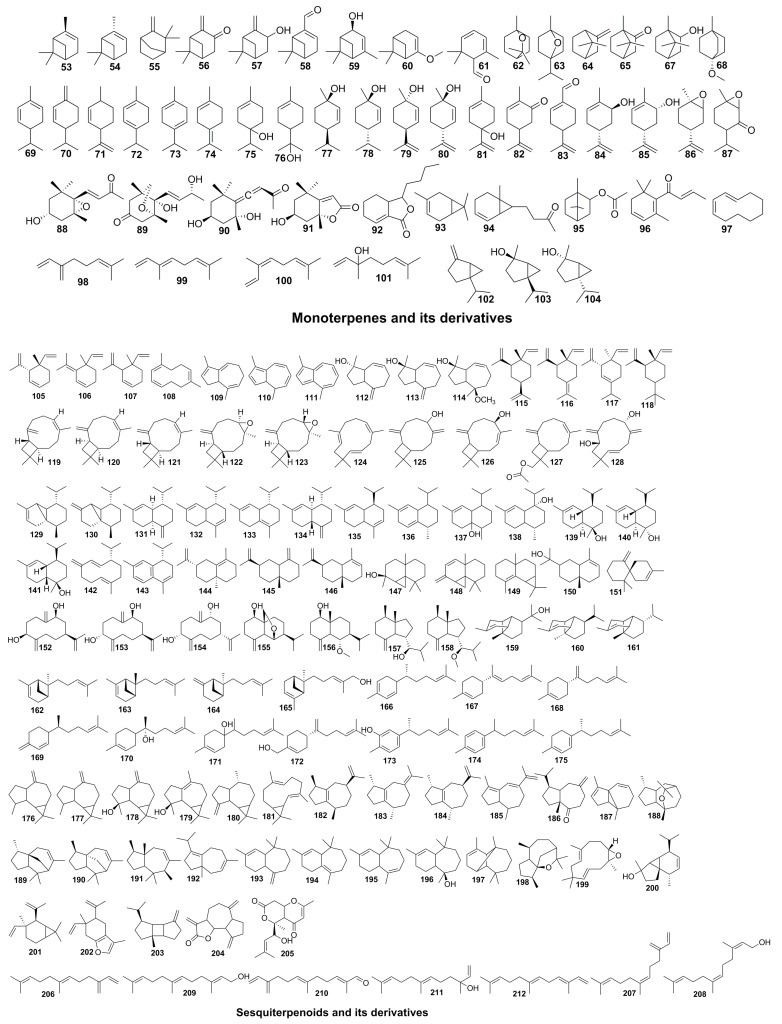
Chemical structures of terpenoids.

**Figure 5 molecules-28-01571-f005:**
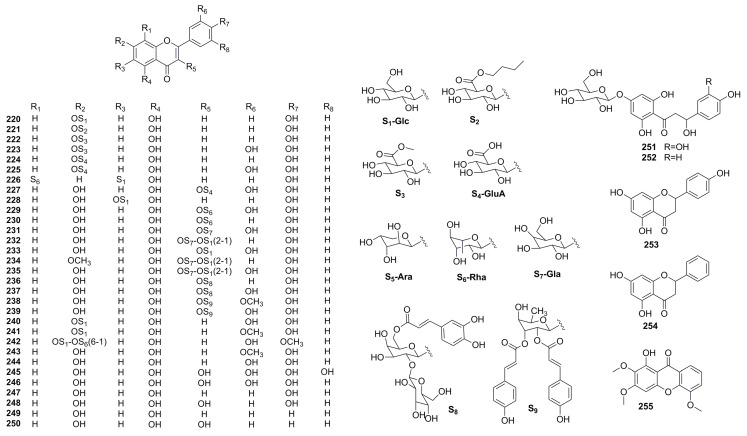
Chemical structures of flavonoidsand their glycosides.

**Figure 6 molecules-28-01571-f006:**
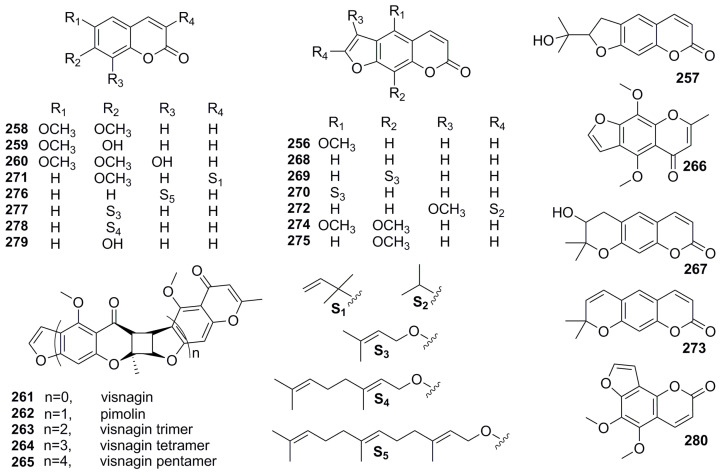
Chemical structures of coumarins.

**Figure 7 molecules-28-01571-f007:**
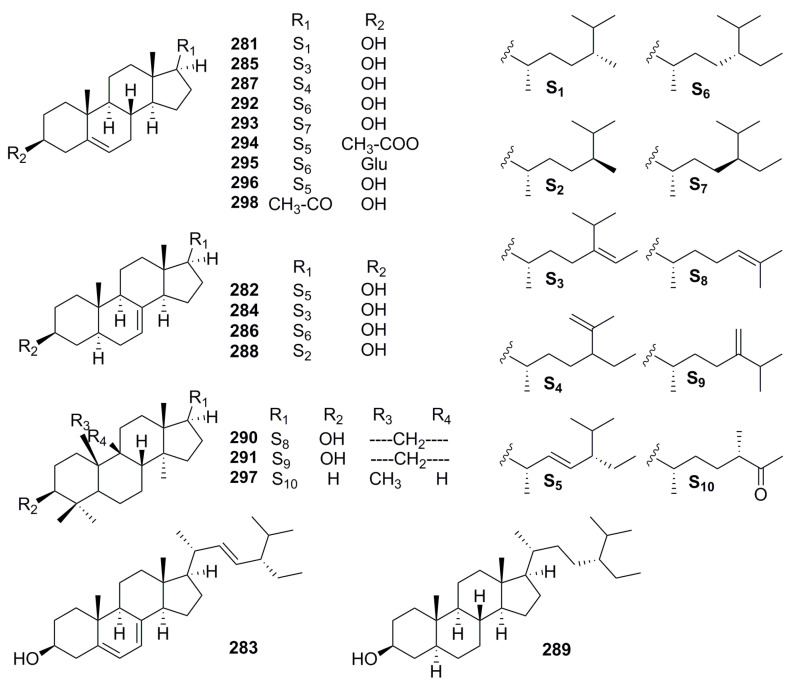
Chemical structures of sterols.

**Figure 8 molecules-28-01571-f008:**
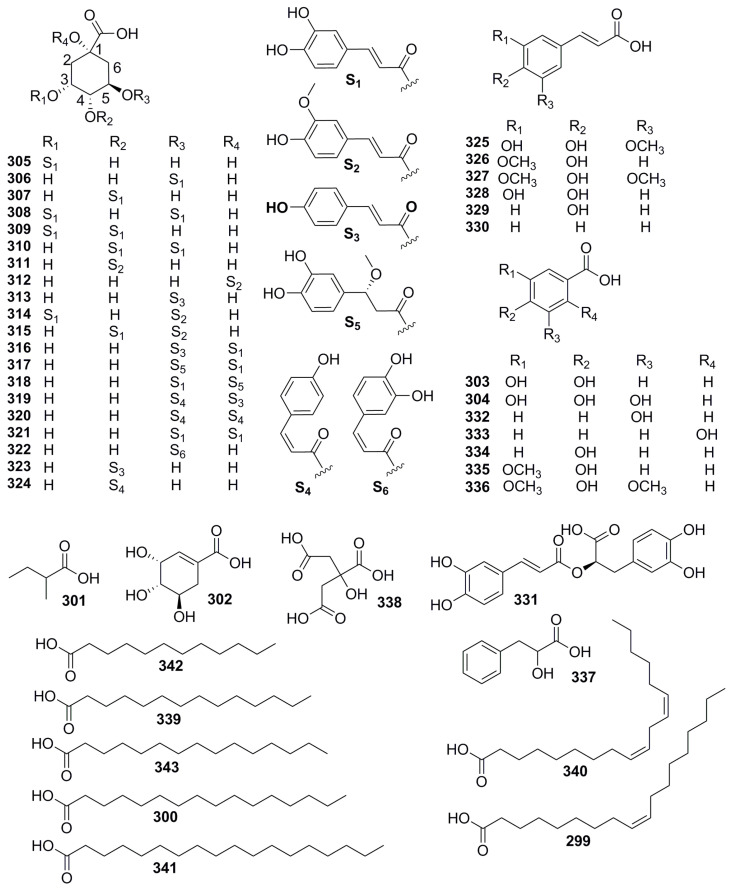
Chemical structures of organic acids.

**Figure 9 molecules-28-01571-f009:**
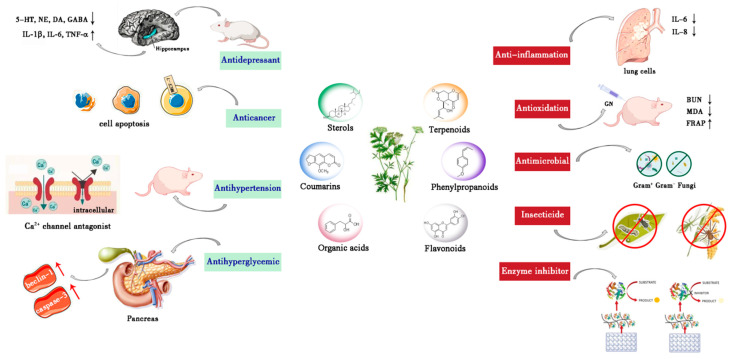
The pharmacological mechanisms of the genus *Pimpinella*.

**Table 1 molecules-28-01571-t001:** The folk-medicine applications of some *Pimpinella* species in several countries.

Part	*Pimpinella* spp.	Folk-Medicine Applications	Country/Region	Reference
Aerial parts	*P. diversifolia*	Cold, dyspepsia, dysentery, and diarrhea	China	[12]
	*P. candollean*	Chest pain, stomach pain, rheumatism, muscle and bone pain, and used as wild vegetables	China	[12]
	*P. thellungiana*	Anticoagulation	Chian	[12]
	*P. brachycarpa*	Gastrointestinal disturbances, bronchial asthma, insomnia, persistent cough, and used as vegetables	Korean	[13] [14]
	*P. cappadocica*	Carminative and digestive	Turkey	[16]
	*P. anisum*	Renal colic, gastrointestinal colic, and upper respiratory tract disease	Egypt	[21]
	*P. anisum*	Renal colic, gastrointestinal colic, and upper respiratory tract disease	Lebanon	[22]
	*P. anisum*	Used as a tea to treat constipation	Brazil	[24]
Seeds	*P. brachycarpa*	Gastrointestinal disturbances, bronchial asthma, insomnia, persistent cough, and used as vegetables	Korean	[14]
	*P. monoica*	Stomachache	India	[15]
	*P. rhodantha*	Sedative, expectorant, and increase lactation	Turkey	[17]
	*P. peregrine*	Carminative, digestive, and increase lactation	Turkey	[18]
	*P. khorasanica*	Carminative, digestive, and increase lactation	Turkey	[19]
	*P. anisum*	Epilepsy	Iran	[20]
	*P. anisum*	Insect repellents, stomach-cramping sedatives, diuretics, and urinary tract disinfectants	England	[23]
	*P. anisum*	Used as plant spice to produce spirits drinks and confectionery	Spain France	[25] [25]
Essential oil	*P. anisum*	Carminative, aromatic, disinfectant, and diuretic	Iran	[20]

**Table 2 molecules-28-01571-t002:** Phenylpropanoids of *Pimpinella* plant.

No.	Name	Formula	Mol. Wt.	Species	Reference
1	2-(1′-ethoxy-2′-hydroxy)propyl-4-methoxyphenol (llungianin A)	C_12_H_18_O_4_	226	*P. thellungiana*	[30]
2	2-(1′-ethoxy-2′-hydroxy)propyl-4-methoxyphenyl-2-methyl-butyrate (llungianin B)	C_17_H_26_O_5_	310	*P. thellungiana*	[30]
3	2-(1′-methoxy-2′-hydroxy) propyl-4-methoxyphenol (llungianin E)	C_11_H_16_O_4_	212	*P. thellungiana*	[31]
4	2-(1′,2′-dihydroxy)propyl-4-methoxyphenol	C_10_H_14_O_4_	198	*P. thellungiana*	[32]
5	4-methoxy-2-propenyl-phenyl-(3′-methyl) butanoate	C_15_H_20_O_3_	248	*P. thellungiana*	[33]
6	2-(1′,2′-epoxy)propyl-4-methoxypheryl-(2″-methyl)-butyrate (llungianin G)	C_15_H_20_O_4_	264	*P. thellungiana* *P. saxifraga*	[34] [46]
7	4-methoxy-2-(3-methyloxiranyl) phenyl-2-methylbutenate	C_15_H_18_O_4_	262	*P. diversifolia* *P. aurea* *P. peregrina*	[35] [45]
8	4-methoxy-2-(3-methyloxiranyl)phenyl isobutyrate	C_14_H_18_O_4_	250	*P. diversifolia* *P. peregrina*	[35] [36]
9	4-methoxy-1-propenyl-phenyl-(2′-methyl) butanoate	C_15_H_20_O_3_	248	*P. anisum*	[29]
10	4-methoxy-2-(1-propenyl)-phenylisobutyrate	C_14_H_18_O_3_	234	*P. peregrina*	[36]
11	4-methoxy-2-(3-methyloxiranyl)-phenylangelate	C_15_H_18_O_4_	262	*P. peregrina*	[36]
12	pseudoisoeugenol	C_10_H_12_O_2_	164	*P. saxifraga*	[37]
13	2-methoxy-4-(3-methyloxiranyl) phenyl 2-methyl butanoate	C_15_H_20_O_4_	264	*P. saxifraga*	[37]
14	2-methoxy-4-(3-methyloxiranyl) phenyl2-methyl butenate	C_15_H_18_O_4_	262	*P. saxifraga*	[38]
15	4-methoxy-2-fromylphenyl-(2′-methyl) butanoate	C_13_H_16_O_4_	236	*P. anisum*	[29]
16	1-angelyloxy-2-(3-methyloxiranyl)-4-isobutyryloxybenzene	C_18_H_22_O_5_	318	*P. diversifolia*	[39]
17	l-isobuty-ryloxy-2-(3-methyloxiranyl)-4-angelyloxybenzene	C_18_H_22_O_5_	318	*P. diversifolia*	[39]
18	1,4-diangelyoxy-2-(3-methyloxiranyl)benzene	C_19_H_22_O_5_	330	*P. diversifolia*	[39]
19	4-propenyl-phenyl-2-methyl butanoate (llungianin F)	C_15_H_20_O_2_	232	*P. thellungiana*	[33]
20	4-(2-methyl-2-butenoyl)oxy)-2-(3-methyloxiran-2-yl)-phenyl2-methyl-2,3-epoxybutanoate	C_19_H_22_O_6_	346	*P. villosa*	[40]
21	4-(2-methyl-2-butenoyloxy)-2-(3-methyloxiran-2-yl)-phenyl 2-methyl-2-butenoate	C_15_H_18_O_5_	278	*P. villosa*	[40]
22	2-methoxy-4-prop-1-enylphenyl isobutyrate	C_14_H_18_O_3_	234	*P. junoniae* *P. aurea*	[44] [45]
23	pseudoisoeugenyltiglate	C_15_H_18_O_3_	246	*P. junoniae*	[44]
24	4-(1-propenyl)-phenyl tiglate	C_14_H_16_O_2_	216	*P. aurea*	[45]
25	4-(1-propenyl)-phenylisobutyrate	C_13_H_16_O_2_	204	*P. corymbosa*	[36]
26	4-(3-methyloxiranyl)-phenyl-2-methylbutyrate	C_14_H_18_O_3_	234	*P. aurea*	[45]
27	4-(3-Methyloxiranyl)-phenyltiglate	C_14_H_16_O_3_	232	*P. aurea*	[45]
28	epoxypseudoisoeugenyl-2-methyl butyrate	C_14_H_18_O_4_	250	*P. corymbosa* *P. peregrina* *P. puberula*	[36]
29	5-(1′-ethoxy-2′-hydroxy)propyl-3-methoxyphenol	C_12_H_18_O_4_	226	*P. thellungiana*	[41]
30	5-methoxy-2-methyl benzofuran (llungianin H)	C_10_H_10_O_2_	162	*P. thellungiana* *P. junoniae* *P. peregrina*	[42] [44] [36]
31	2-methyl-2-hydroxy-5-methoxy berzo (d) hydrofuran-3-one	C_10_H_10_O_4_	194	*P. thellungiana*	[43]
32	*erythro*-1′-(4-methoxyphenyl)-propan-1′,2′-diol	C_10_H_14_O_3_	182	*P. aurea*	[45]
33	*erythro*-1′-[4-(*sec*-butyl)-phenyl]-propan-1′,2′-diol	C_13_H_20_O_2_	208	*P. aurea*	[45]
34	eugenol	C_10_H_12_O_2_	164	*P. puberula*	[36]
35	elemicine	C_12_H_16_O_3_	208	*P. puberula*	[36]
36	*p*-cymene	C_10_H_14_	134	*P. anisetum* *P. aurea* *P. corymbosa*	[48] [45]
37	*α*,*p*-dimethylstyrene	C_10_H_12_	132	*P. aurea*	[45]
38	1-(4-hydroxyphenyl)-1,2-ethanediol	C_8_H_10_O_3_	154	*P. candolleana*	[49]
39	methyl chavicol	C_10_H_12_O	148	*P. anisetum* *P. anisum*	[48] [50]
40	*cis*-anethole	C_10_H_12_O	148	*P. anisetum* *P. flabellifolia* *P. saxifrage* *P. anisum*	[48] [46] [50]
41	4-propenylphenol	C_9_H_10_O	134	*P. thellungiana*	[41]
42	*trans*-anethole	C_10_H_12_O	148	*P. anisetum* *P. flabellifolia* *P. aurea* *P. corymbosa* *P. peregrine* *P. anisum*	[48] [36] [50]
43	methyl isoeugenol	C_11_H_14_O_2_	178	*P. flabellifolia*	[48]
44	*p*-cymen-8-ol	C_10_H_14_O	150	*P. junoniae* *P. aurea*	[44] [45]
45	methyl eugenol	C_11_H_14_O_2_	178	*P. corymbosa* *P. puberula*	[36]
46	carvacrol	C_11_H_14_O_2_	178	*P. aurea* *P. corymbosa* *P. puberula*	[36]
47	*p*-anisaldehyde	C_10_H_12_O_2_	164	*P. saxifrage* *P. anisum*	[46] [50]
48	methyl-*O*-coumarate	C_10_H_10_O_3_	178	*P. saxifraga*	[46]
49	1-(2-hydroxy-4-methoxyphenyl)propan1-one	C_10_H_12_O_3_	180	*P. saxifraga*	[46]
50	4-methoxycinnamaldehyde	C_10_H_12_O_3_	180	*P. saxifraga*	[46]
51	dillapiole	C_12_H_14_O_4_	222	*P. saxifrage* *P. serbica*	[46] [47]
52	nothoapiole	C_13_H_16_O_5_	252	*P. serbica*	[47]

**Table 3 molecules-28-01571-t003:** Species of *Pimpinella* plants.

No.	Type	Name	Formula	Mol. Wt.	Species	Reference
53	Monoterpenoid	*α*-pinene	C_10_H_16_	136	*P. aurea* *P. corymbosa* *P. peregrina* *P. puberula* *P. junoniae* *P. anisetum* *P. flabellifolia* *P. anisum* *P. affinis* *P. monoica* *P. thellungiana*	[36] [44] [48] [57] [52] [15] [54]
54	Monoterpenoid	*β*-pinene	C_10_H_16_	136	*P. aurea* *P. corymbosa* *P. puberula* *P. flabellifolia* *P. anisum* *P. monoica* *P. thellungiana*	[36] [48] [6] [15] [54]
55	Monoterpenoid	camphene	C_10_H_16_	136	*P. aurea* *P. corymbosa* *P. flabellifolia*	[36] [48]
56	Monoterpenoid	pinocarvone	C_10_H_14_O	150	*P. aurea*	[36]
57	Monoterpenoid	pinocarveol	C_10_H_16_O	152	*P. aurea* *P. thellungiana*	[36] [54]
58	Monoterpenoid	myrtenal	C_10_H_14_O	150	*P. corymbosa* *P. thellungiana*	[36] [54]
59	Monoterpenoid	*trans*-verbenol	C_10_H_16_O	152	*P. corymbosa* *P. peregrine* *P. monoica*	[36] [15]
60	Monoterpenoid	myrtenol	C_10_H_16_O	152	*P. aurea*	[36]
61	Monoterpenoid	safranal	C_10_H_14_O	150	*P. anisum*	[57]
62	Monoterpenoid	1,8-cineole	C_10_H_18_O	154	*P. anisum*	[57]
63	Monoterpenoid	1,4-cineole	C_10_H_18_O	154	*P. thellungiana*	[54]
64	Monoterpenoid	*α*-fenchene	C_10_H_16_	136	*P. monoica*	[15]
65	Monoterpenoid	camphor	C_10_H_16_O	152	*P. anisum*	[58]
67	Monoterpenoid	borneol	C_10_H_18_O	154	*P. anisum* *P. monoica*	[58] [15]
68	Monoterpenoid	1-methoxy-4-methylbicyclo[2.2.2]octane	C_10_H_18_O	154	*P. thellungiana*	[54]
69	Monoterpenoid	*α*-phellandrene	C_10_H_16_	136	*P. flabellifolia* *P. anisum*	[48] [57]
70	Monoterpenoid	*β*-phellandrene	C_10_H_16_	136	*P. aurea* *P. puberula* *P. junoniae* *P. anisum*	[36] [44] [58]
71	Monoterpenoid	Limonene	C_10_H_16_	136	*P. aurea* *P. corymbosa* *P. puberula* *P. anisetum* *P. flabellifolia* *P. anisum* *P. enguezekensis* *P. affinis* *P. monoica*	[36] [48] [57] [51] [52] [15]
72	Monoterpenoid	*α*-terpinene	C_10_H_16_	136	*P. aurea* *P. puberula* *P. anisum* *P. monoica*	[36] [50] [15]
73	Monoterpenoid	*γ*-terpinene	C_10_H_16_	136	*P. aurea* *P. flabellifolia* *P. anisum* *P. enguezekensis* *P. monoica*	[36] [48] [6] [51] [15]
74	Monoterpenoid	Terpinolene	C_10_H_16_	136	*P. aurea* *P. junoniae* *P. anisum* *P. monoica*	[36] [44] [58] [15]
75	Monoterpenoid	terpinen-4-ol	C_10_H_18_O	154	*P. aurea* *P. junoniae* *P. flabellifolia* *P. anisum*	[36] [44] [48] [57]
76	Monoterpenoid	*α*-terpineol	C_10_H_18_O	154	*P. junoniae* *P. flabellifolia* *P. anisum* *P. monoica*	[44] [48] [57] [15]
77	Monoterpenoid	*trans-p*-menth-2-en-1-ol	C_10_H_18_O	154	*P. aurea*	[36]
78	Monoterpenoid	*cis-p*-menth-2-en-1-ol	C_10_H_18_O	154	*P. aurea*	[36]
79	Monoterpenoid	*trans-p*-mentha-2,8-dien-1-ol	C_10_H_16_O	152	*P. puberula* *P. flabellifolia*	[36] [48]
80	Monoterpenoid	*cis-p-*mentha-2,8-dien-1-ol	C_10_H_16_O	152	*P. puberula* *P. flabellifolia*	[36] [48]
81	Monoterpenoid	*p-*mentha-1,8-dien-4-ol	C_10_H_16_O	152	*P. aurea*	[36]
82	Monoterpenoid	carvone	C_10_H_14_O	150	*P. puberula* *P. anisum* *P. enguezekensis*	[36] [6] [51]
83	Monoterpenoid	perilla aldehyde	C_10_H_14_O	150	*P. puberula*	[36]
84	Monoterpenoid	*trans*-carveol	C_10_H_16_O	152	*P. puberula* *P. anisum*	[36] [57]
85	Monoterpenoid	*cis*-carveol	C_10_H_16_O	152	*P. anisum*	[57]
86	Monoterpenoid	*cis*-1,2-limonene epoxide	C_10_H_16_O	152	*P. puberula*	[36]
87	Monoterpenoid	piperitone oxide	C_10_H_16_O_2_	168	*P. thellungiana*	[54]
88	Monoterpenoid	3-hydroxy-5,6-epoxy-7-megastigmen-9-one	C_13_H_20_O_3_	224	*P. brachycarpa*	[13]
89	Monoterpenoid	(1*R*,6*R*,9*R*)-6,9,11-trihydroxy-4-megastigmen-3-one	C_13_H_20_O_4_	240	*P. brachycarpa*	[13]
90	Monoterpenoid	grasshopper ketone	C_13_H_20_O_3_	224	*P. brachycarpa*	[13]
91	Monoterpenoid	loliolide	C_11_H_16_O_3_	196	*P. brachycarpa*	[13]
92	Monoterpenoid	sedanolide	C_12_H_18_O_2_	194	*P. puberula*	[36]
93	Monoterpenoid	*δ*-3-carene	C_10_H_16_	136	*P. aurea* *P. corymbosa* *P. puberula* *P. anisum* *P. enguezekensis*	[36] [59] [51]
94	Monoterpenoid	traginone	C_12_H_18_O	178	*P. puberula*	[36]
95	Monoterpenoid	bornyl acetate	C_12_H_20_O_2_	196	*P. aurea* *P. puberula*	[36]
96	Monoterpenoid	*trans*-*β*-damascenone	C_13_H_18_O	190	*P. puberula*	[36]
97	Monoterpenoid	cyclodecadiene	C_10_H_16_	136	*P. diversifolia*	[53]
98	Monoterpenoid	*β*-myrcene	C_10_H_16_	136	*P. aurea* *P. corymbosa* *P. puberula* *P. anisetum* *P. flabellifolia* *P. anisum* *P. affinis* *P. monoica*	[36] [48] [59] [52] [15]
99	Monoterpenoid	*trans*-*β*-ocimene	C_10_H_16_	136	*P. aurea* *P. anisum* *P. monoica*	[36] [58] [15]
100	Monoterpenoid	*cis*-*β*-ocimene	C_10_H_16_	136	*P. anisum* *P. affinis* *P. monoica*	[58] [52] [15]
101	Monoterpenoid	Linalool	C_10_H_18_O	154	*P. junoniae* *P. flabellifolia* *P. anisum* *P. enguezekensis* *P. affinis* *P. diversifolia*	[44] [48] [50] [51] [52] [53]
102	Monoterpenoid	sabinene	C_10_H_16_	136	*P. aurea* *P. corymbosa* *P. puberula* *P. flabellifolia* *P. anisum* *P. monoica*	[36] [48] [57] [15]
103	Monoterpenoid	*trans*-sabinene hydrate	C_10_H_18_O	154	*P. aurea*	[36]
104	Monoterpenoid	*cis*-sabinene hydrate	C_10_H_18_O	154	*P. aurea*	[36]
105	C_12_-sesquiterpenes	isogeijerene	C_12_H_18_	162	*P. corymbosa* *P. puberula*	[36]
106	C_12_-sesquiterpenes	isogeijerene C	C_12_H_18_	162	*P. puberula*	[36]
107	C_12_-sesquiterpenes	geijerene	C_12_H_18_	152	*P. aurea* *P. corymbosa* *P. peregrina* *P. puberula* *P. anisetum* *P. anisum* *P. affinis* *P. khorasanica* *P. thellungiana*	[36] [48] [50] [52] [19] [54]
108	C_12_-sesquiterpenes	pregeijerene	C_12_H_18_	162	*P. corymbosa* *P. puberula* *P. affinis* *P. khorasanica*	[36] [52] [19]
109	C_12_-sesquiterpenes	3,10-dihydro-1,4-dimethylazulene	C_12_H_14_	158	*P. puberula*	[36]
110	C_12_-sesquiterpenes	4,10-dihydro-1,4-dimethylazulene	C_12_H_14_	158	*P. corymbosa*	[36]
111	C_12_-sesquiterpenes	1,4-dimethylazulene	C_12_H_12_	156	*P. corymbosa*	[36]
112	C_12_-sesquiterpenes	8-*epi*-dictamnol	C_12_H_18_O	178	*P. puberula*	[36]
113	C_12_-sesquiterpenes	dictamnol	C_12_H_18_O	178	*P. puberula* *P. affinis*	[36] [52]
114	C_12_-sesquiterpenes	1*α*, 5*α*-dimethyl-4*α*, 10*α*-bicyclo [0,3,5] dec-8-en-5*β*-methoxy-1*β*-ol	C_13_H_22_O_2_	210	*P. cappadocica*	[16]
115	Sesquiterpenes	*β*-elemene	C_15_H_24_	204	*P. aurea* *P. corymbosa* *P. anisum* *P. diversifolia*	[36] [50] [53]
116	Sesquiterpenes	*γ*-elemene	C_15_H_24_	204	*P. flabellifolia* *P. monoica*	[48] [15]
117	Sesquiterpenes	*δ*-elemene	C_15_H_24_	204	*P. corymbosa* *P. anisum* *P. enguezekensis* *P. affinis*	[36] [50] [51] [52]
118	Sesquiterpenes	elemol	C_15_H_26_O	222	*P. puberula*	[36]
119	Sesquiterpenes	*β*-caryophyllene	C_15_H_24_	204	*P. aurea* *P. corymbosa* *P. peregrina* *P. puberula* *P. anisetum* *P. anisum* *P. monoica* *P. diversifolia*	[36] [48] [57] [15] [53]
120	Sesquiterpenes	9-*epi*-β-caryophyllene	C_15_H_24_	204	*P. peregrina*	[36]
121	Sesquiterpenes	isocaryophyllene	C_15_H_24_	204	*P. peregrina*	[36]
122	Sesquiterpenes	isocaryophyllene oxide	C_15_H_24_O	220	*P. corymbosa* *P. peregrina*	[36]
123	Sesquiterpenes	caryophyllene oxide	C_15_H_24_O	220	*P. aurea* *P. corymbosa* *P. peregrina* *P. puberula* *P. monoica* *P. diversifolia* *P. thellungiana*	[36] [15] [53] [54]
124	Sesquiterpenes	*α*-humulene	C_15_H_24_	204	*P. corymbosa* *P. peregrine* *P. monoica* *P. diversifolia*	[36] [15] [53]
125	Sesquiterpenes	caryophylladienol II	C_15_H_24_O	220	*P. peregrine*	[36]
126	Sesquiterpenes	caryophyllenol II	C_15_H_24_O	220	*P. puberula*	[36]
127	Sesquiterpenes	12-hydroxy-*β*-caryophylleneacetate	C_17_H_26_O_2_	262	*P. aurea* *P. corymbosa*	[36]
128	Sesquiterpenes	(2*R**,6*S**)-2,6-dihydroxyhumlaobtusa	C_15_H_24_O_2_	236	*P. brachycarpa*	[13]
129	Sesquiterpenes	*α*-cubebene	C_15_H_24_	204	*P. corymbosa* *P. junoniae* *P. monoica*	[36] [44] [15]
130	Sesquiterpenes	*β*-cubebene	C_15_H_24_	204	*P. corymbosa* *P. junoniae* *P. affinis* *P. monoica* *P. diversifolia*	[36] [44] [52] [15] [53]
131	Sesquiterpenes	*γ*-muurolene	C_15_H_24_	204	*P. aurea* *P. corymbosa* *P. peregrine* *P. junoniae* *P. enguezekensis*	[36] [44] [51]
132	Sesquiterpenes	*α*-cadinene	C_15_H_24_	204	*P. corymbosa*	[36]
133	Sesquiterpenes	*δ*-cadinene	C_15_H_24_	204	*P. corymbosa* *P. anisetum* *P. anisum* *P. monoica* *P. diversifolia*	[36] [48] [58] [15] [53]
134	Sesquiterpenes	*γ*-cadinene	C_15_H_24_	204	*P. junoniae* *P. monoica*	[44] [15]
135	Sesquiterpenes	*α*-amorphene	C_15_H_24_	204	*P. aurea*	[45]
136	Sesquiterpenes	cadina-1,4-diene	C_15_H_24_	204	*P. corymbosa*	[36]
137	Sesquiterpenes	1-*epi*-cubenol	C_15_H_26_O	222	*P. corymbosa*	[36]
138	Sesquiterpenes	*cis*-cadin-4-en-7-ol	C_15_H_26_O	222	*P. aurea*	[45]
139	Sesquiterpenes	T-cadinol	C_15_H_26_O	222	*P. corymbosa*	[36]
140	Sesquiterpenes	*α*-cadinol	C_15_H_26_O	222	*P. corymbosa* *P. anisum*	[36] [59]
141	Sesquiterpenes	T-muurolol	C_15_H_26_O	222	*P. corymbosa*	[36]
142	Sesquiterpenes	germacrene D	C_15_H_24_	204	*P. aurea* *P. corymbosa* *P. peregrina* *P. puberula* *P. anisetum* *P. anisum* *P. enguezekensis* *P. affinis* *P. monoica* *P. thellungiana*	[36] [48] [50] [51] [52] [15] [54]
143	Sesquiterpenes	*α*-calacorene	C_15_H_20_	200	*P. corymbosa* *P. anisum* *P. monoica*	[36] [58] [15]
144	Sesquiterpenes	4,11-selinadiene	C_15_H_24_	204	*P. saxifraga*	[46]
145	Sesquiterpenes	*β*-selinene	C_15_H_24_	204	*P. saxifrage* *P. anisum*	[46] [57]
146	Sesquiterpenes	*α*-selinene	C_15_H_24_	204	*P. monoica*	[15]
147	Sesquiterpenes	thujopsan-2-*α*-ol	C_15_H_26_O	222	*P. aurea*	[45]
148	Sesquiterpenes	Thujpsadiene	C_15_H_22_	202	*P. saxifraga*	[46]
149	Sesquiterpenes	cyclopropa[a]naphthalene	C_15_H_24_	204	*P. diversifolia*	[53]
150	Sesquiterpenes	7-*epi*-*α*-eudesmol	C_15_H_26_O	222	*P. aurea*	[45]
151	Sesquiterpenes	*β*-chamigrene	C_15_H_24_	204	*P. anisum* *P. diversifolia*	[57] [53]
152	Sesquiterpenes	(3*S*,7*S*,9*S*)-3,9-dihydroxygermacra-4(15),10(14),11(12)-triene	C_15_H_24_O_2_	236	*P. brachycarpa*	[13]
153	Sesquiterpenes	(3*R*,7*S*,9*S*)-3,9-dihydroxygermacra-4(15),10(14),11(12)-triene	C_15_H_24_O_2_	236	*P. brachycarpa*	[13]
154	Sesquiterpenes	(3*R*,7*R*,9*R*)-3,9-dihydroxygermacra-4(15),10(14),11(12)-triene	C_15_H_24_O_2_	236	*P. brachycarpa*	[13]
155	Sesquiterpenes	6*β*,14-epoxyeudesm-4(15)-en-1*β*-ol	C_15_H_24_O_2_	236	*P. brachycarpa*	[13]
156	Sesquiterpenes	6*β*-methoxyeudesm-4(15)-en-1*β*-ol	C_16_H_28_O_2_	252	*P. brachycarpa*	[13]
157	Sesquiterpenes	(7R*)-opposit-4(15)-ene-1*β*,7-diol	C_16_H_28_O	236	*P. brachycarpa*	[13]
158	Sesquiterpenes	7*β*-methoxy-4(14)-oppositen-1*β*-ol	C_17_H_30_O	250	*P. brachycarpa*	[13]
159	Sesquiterpenes	*α*-copaene-11-ol	C_15_H_24_O	220	*P. corymbosa*	[36]
160	Sesquiterpenes	*α-*ylangene	C_15_H_24_	204	*P. anisetum* *P. anisum*	[48] [58]
161	Sesquiterpenes	*α*-copaene	C_15_H_24_	204	*P. aurea* *P. corymbosa* *P. peregrine* *P. junoniae* *P. anisum* *P. monoica* *P. thellungiana*	[36] [44] [58] [15] [54]
162	Sesquiterpenes	*trans*-*α*-bergamotene	C_15_H_24_	204	*P. peregrine* *P. junoniae* *P. anisum*	[36] [44] [59]
163	Sesquiterpenes	*cis*-*α*-bergamotene	C_15_H_24_	204	*P. peregrina*	[36]
164	Sesquiterpenes	*trans*-*β*-bergamotene	C_15_H_24_	204	*P. peregrina*	[36]
165	Sesquiterpenes	*trans*-*α*-bergamotol	C_15_H_24_O	220	*P. corymbosa*	[36]
166	Sesquiterpenes	*α*-zingiberene	C_15_H_24_	204	*P. corymbosa* *P. junoniae* *P. anisetum* *P. anisum* *P. enguezekensis* *P. khorasanica* *P. diversifolia*	[36] [44] [48] [50] [51] [19] [53]
167	Sesquiterpenes	*trans*-*α*-bisabolene	C_15_H_24_	204	*P. corymbosa*	[36]
168	Sesquiterpenes	*β*-bisabolene	C_15_H_24_	204	*P. corymbosa* *P. peregrina* *P. puberula* *P. junoniae* *P. aurea* *P. anisetum* *P. anisum* *P. enguezekensis* *P. khorasanica* *P. diversifolia* *P. thellungiana*	[36] [44] [45] [48] [50] [51] [19] [53] [54]
169	Sesquiterpenes	*β*-sesquiphellandrene	C_15_H_24_	204	*P. peregrine* *P. junoniae* *P. diversifolia* *P. thellungiana*	[36] [44] [53] [54]
170	Sesquiterpenes	*α*-bisabolol	C_15_H_26_O	222	*P. aurea* *P. corymbosa* *P. peregrine* *P. junoniae* *P. thellungiana*	[36] [44] [54]
171	Sesquiterpenes	*β*-bisabolol	C_15_H_26_O	222	*P. aurea*	[45]
172	Sesquiterpenes	*β*-bisabolenol	C_15_H_24_O	220	*P. aurea*	[36]
173	Sesquiterpenes	xanthorrhizol	C_15_H_22_O	218	*P. junoniae*	[44]
174	Sesquiterpenes	*α*-curcumene	C_15_H_22_	202	*P. junoniae* *P. anisum* *P. khorasanica* *P. thellungiana*	[44] [57] [19] [54]
175	Sesquiterpenes	*γ*-curcumene	C_15_H_24_	202	*P. thellungiana*	[54]
176	Sesquiterpenes	dehydro aromadendrene	C_15_H_22_	204	*P. monoica*	[15]
177	Sesquiterpenes	aromadendrene	C_15_H_24_	204	*P. anisetum* *P. diversifolia*	[48] [53]
178	Sesquiterpenes	spathulenol	C_15_H_24_O	220	*P. corymbosa* *P. junoniae* *P. aurea* *P. anisetum* *P. thellungiana*	[36] [44] [45] [48] [54]
179	Sesquiterpenes	isospathulenol	C_15_H_24_O	220	*P. thellungiana*	[54]
180	Sesquiterpenes	*β*-gurjunene	C_15_H_24_	204	*P. junoniae*	[44]
181	Sesquiterpenes	bicyclogermacrene	C_15_H_24_	204	*P. aurea* *P. corymbosa* *P. peregrina* *P. flabellifolia*	[36] [48]
182	Sesquiterpenes	*α*-guaiene	C_15_H_24_	204	*P. diversifolia*	[54]
183	Sesquiterpenes	*cis*-*β*-guaiene	C_15_H_24_	204	*P. junoniae*	[44]
184	Sesquiterpenes	*trans*-*β*-guaiene	C_15_H_24_	204	*P. junoniae*	[44]
185	Sesquiterpenes	4,6-guaiadiene	C_15_H_22_	202	*P. corymbosa* *P. peregrina*	[36]
186	Sesquiterpenes	salvial-4(14)-en-1-one	C_15_H_24_O	220	*P. thellungiana*	[54]
187	Sesquiterpenes	clavukerin B	C_12_H_16_	160	*P. corymbosa*	[36]
188	Sesquiterpenes	kessane	C_15_H_26_O	222	*P. aurea*	[36]
189	Sesquiterpenes	*α*-cedrene	C_15_H_24_	204	*P. monoica* *P. diversifolia*	[15] [53]
190	Sesquiterpenes	2-*epi*-*α*-funebrene	C_15_H_24_	204	*P. monoica*	[15]
191	Sesquiterpenes	diepi-*α*-cedrene	C_16_H_28_	220	*P. anisum*	[50]
192	Sesquiterpenes	daucene	C_15_H_24_	204	*P. monoica*	[15]
193	Sesquiterpenes	*α*-himachalene	C_15_H_24_	204	*P. corymbosa* *P. anisum* *P. enguezekensis*	[36] [50] [51]
194	Sesquiterpenes	*β*-himachalene	C_15_H_24_	204	*P. anisum*	[50]
195	Sesquiterpenes	*γ*-himachalene	C_15_H_24_	204	*P. corymbosa* *P. anisetum* *P. anisum*	[36] [48] [50]
196	Sesquiterpenes	himachalol	C_15_H_26_O	222	*P. corymbosa* *P. peregrina* *P. aurea*	[36] [45]
197	Sesquiterpenes	*α*-longipinene	C_15_H_24_	204	*P. anisum* *P. thellungiana*	[58] [54]
198	Sesquiterpenes	guaioxide	C_15_H_26_O	222	*P. aurea*	[36]
199	Sesquiterpenes	humulene epoxide II	C_15_H_24_O	220	*P. peregrina*	[36]
200	Sesquiterpenes	*epi*-cubebol	C_15_H_24_O	220	*P. corymbosa*	[36]
201	Sesquiterpenes	bicycloelemene	C_15_H_24_	204	*P. aurea* *P. corymbosa*	[36]
202	Sesquiterpenes	isofuranogermacrene	C_15_H_20_O	204	*P. diversifolia*	[53]
203	Sesquiterpenes	*β*-bourbonene	C_15_H_24_	204	*P. corymbosa* *P. junoniae* *P. anisum*	[36] [44] [58]
204	Sesquiterpenes	dehydrocostus lactone	C_15_H_18_O_2_	230	*P. puberula*	[36]
205	Sesquiterpenes	Pimpinelol	C_15_H_20_O_5_	280	*P. haussknechtii*	[10]
206	Sesquiterpenes	*trans*-*β*-farnesene	C_15_H_24_	204	*P. peregrine* *P. junoniae* *P. khorasanica* *P. diversifolia*	[36] [44] [19] [53]
207	Sesquiterpenes	*cis*-*β*-farnesene	C_15_H_24_	204	*P. aurea* *P. corymbosa* *P. peregrine* *P. thellungiana*	[36] [54]
208	Sesquiterpenes	*cis, cis*-farnesol	C_15_H_26_O	222	*P. junoniae*	[44]
209	Sesquiterpenes	*trans, trans*-farnesol	C_15_H_26_O	222	*P. junoniae*	[44]
210	Sesquiterpenes	sinensal	C_15_H_22_O	218	*P. peregrina*	[36]
211	Sesquiterpenes	nerolidol	C_15_H_26_O	222	*P. anisum* *P. diversifolia*	[59] [53]
212	Sesquiterpenes	*cis,trans-α*-farnesene	C_15_H_24_	204	*P. thellungiana*	[54]
213	Triterpenoids	ursolic acid	C_30_H_48_O_3_	456	*P. anisum*	[56]
214	Triterpenoids	oleanolic acid	C_30_H_48_O_3_	456	*P. anisum*	[56]
215	Triterpenoids	betulinic acid	C_30_H_48_O_3_	456	*P. anisum*	[56]
216	Triterpenoids	lupeol	C_30_H_50_O	426	*P. anisum*	[56]
217	Triterpenoids	*α*-amyrin	C_30_H_50_O	426	*P. anisum*	[55]
218	Triterpenoids	*β*-amyrin	C_30_H_50_O	426	*P. anisum*	[55]
219	Triterpenoids	saikogenin F-3-*O*-{β-D- glucopyranosyl-(1→2)-[β-D-xylopyranosyl-(1→4)-β-D-glucopyranosyl-(1→3)]-β-D-fucopyranoside}	C_53_H_86_O_22_	1075	*P. rhodantha*	[17]

**Table 4 molecules-28-01571-t004:** Flavonoids and their glycosides of *Pimpinella* plants.

No.	Name	Formula	Mol. Wt.	Species	Reference
220	apigenin-7-*O*-glucoside	C_21_H_20_O_10_	432	*P. thellungiana*	[60]
221	apigenin-7-*O*-*β*-D-butylglucuronide	C_25_H_26_O_10_	502	*P. thellungiana*	[60]
222	apigenin-7-*O*-methylglucuronide	C_20_H_22_O_11_	460	*P. thellungiana*	[61]
223	luteolin-7-*O*-methylglucuronide	C_22_H_20_O_12_	476	*P. thellungiana*	[61]
224	apigenin-7-*O*-glucuronide	C_21_H_18_O_11_	446	*P. thellungiana*	[62]
225	luteolin-7-*O*-glucuronide	C_21_H_18_O_12_	462	*P. thellungiana*	[62]
226	schaftoside	C_26_H_28_O_14_	564	*P. thellungiana*	[63]
227	quercetin-3-*O*-glucuronide	C_21_H_18_O_13_	478	*P. thellungiana*	[64]
228	isovitexin	C_21_H_20_O_10_	432	*P. candolleana*	[49]
229	quercetin-3-*O*-rhamnoside	C_21_H_20_O_11_	448	*P. brachycarpa*	[12]
230	kaempferol-3-*O*-rhamnoside	C_22_H_22_O_10_	446	*P. cappadocica*	[16]
231	quercetin-3-*O*-galactoside	C_21_H_20_O_12_	464	*P. cappadocica*	[16]
232	kaempferol-3-*O*-(2″-*O*-glucopyranosyl)-galactoside	C_27_H_30_O_16_	610	*P. cappadocica*	[16]
233	quercetin-3-*O*-glucoside	C_21_H_20_O_12_	464	*P. cappadocica*	[16]
234	rhamnositrin-3-*O*-(2″-*O*-glucopyranosyl)-galactoside	C_28_H_30_O_18_	624	*P. cappadocica*	[16]
235	quercetin-3-*O*-(2″-*O*-glucopyranosyl)-galactoside	C_27_H_30_O_17_	626	*P. cappadocica*	[16]
236	kaempferol-3-*O*-(2″-*O*-*β*-D-glucopyranosyl-6″-*O*-caffeoyl)-*β*-D-galactopyranoside (erzurumin)	C_36_H_36_O_19_	772	*P. cappadocica*	[16]
237	quercetin-3-*O*-(2″-*O*-*β*-D-glucopyranosyl-6″-*O*-caffeoyl)-*β*-D-galactopyranoside (ilicanin)	C_36_H_36_O_20_	788	*P. cappadocica*	[16]
238	quercetin-3′-methylether-3-*O*-*α*-L-(2″,3″-di-*O-trans*-coumaroyl)-rhamnopyranoside	C_40_H_34_O_15_	754	*P. rhodantha*	[17]
239	quercetin-3-*O*-*α*-L-(2″,3″-di-*O-trans*-coumaroyl)-rhamnopyranoside	C_39_H_34_O_13_	740	*P. rhodantha*	[17]
240	luteolin-7-*O*-glucoside	C_21_H_20_O_11_	302	*P. anthriscoides*	[65]
241	chrysoeriol-7-*O*-glucoside	C_22_H_22_O_11_	462	*P. anthriscoides*	[65]
242	diosmetin-7-O-rutinoside	C_28_H_32_O_15_	608	*P. anthriscoides*	[65]
243	chrysoeriol	C_16_H_12_O_6_	300	*P. anthriscoides*	[65]
244	luteolin	C_15_H_10_O_6_	286	*P. candolleana*	[49]
245	myricetin	C_15_H_10_O_8_	318	*P. anisum*	[66]
246	quercetin	C_15_H_10_O_7_	302	*P. anisum*	[66]
247	apigenin	C_15_H_10_O_5_	270	*P. anisum*	[66]
248	kaempferol	C_15_H_10_O_6_	286	*P. anisum*	[66]
249	chrysin	C_15_H_10_O_4_	254	*P. anisum*	[66]
250	galangin	C_15_H_10_O_5_	270	*P. anisum*	[66]
251	(*βR*)-*β*, 3, 4, 2′, 6′ -pentahydroxy-4′-*O*-*β*-D-glucosyldihydrochalcone (ziganin)	C_21_H_24_O_12_	468	*P. rhodantha*	[17]
252	3-hydroxy-*p*-phlorizin	C_21_H_24_O_11_	452	*P. rhodantha*	[17]
253	naringenin	C_15_H_12_O_5_	272	*P. anisum*	[66]
254	pinocembrin	C_15_H_12_O_4_	256	*P. anisum*	[66]
255	1-hydroxy-2, 3, 5-trimethoxyxathone	C_16_H_14_O_6_	302	*P. candolleana*	[49]

**Table 5 molecules-28-01571-t005:** Species of *Pimpinella* plants.

No.	Name	Formula	Mol. Wt.	Species	Reference
256	bergapten	C_12_H_8_O_4_	216	*P. thellungiana*	[67]
257	marmesin	C_14_H_4_O_4_	246	*P. thellungiana*	[67]
258	scoparone	C_11_H_10_O_4_	206	*P. thellungiana*	[67]
259	scopoletin	C_10_H_8_O_4_	192	*P. thellungiana*	[67]
260	isofraxidin	C_11_H_10_O_5_	222	*P. thellungiana*	[67]
261	visnagin	C_13_H_10_O_4_	230	*P. monoica*	[69]
262	pimolin	C_26_H_20_O_8_	460	*P. monoica*	[68]
263	visnagintrimer	C_39_H_30_O_12_	690	*P. monoica*	[69]
264	visnagin tetramer	C_52_H_40_O_16_	920	*P. monoica*	[69]
265	visnagin pentamer	C_65_H_50_O_20_	1150	*P. monoica*	[69]
266	khellin	C_14_H_12_O_5_	260	*P. monoica*	[69]
267	aegelinol	C_14_H_14_O_4_	246	*P. anthriscoides*	[65]
268	psoralen	C_11_H_6_O_3_	186	*P. anthriscoides*	[65]
269	imperatorin	C_16_H_14_O_4_	270	*P. anthriscoides*	[65]
270	isoimperatorin	C_16_H_14_O_4_	270	*P. anthriscoides*	[65]
271	3-(1, 1-dimethylallyl) herniarin	C_15_H_16_O_3_	228	*P. anthriscoides*	[65]
272	peucedanin	C_15_H_14_O_4_	258	*P. anthriscoides*	[65]
273	xanthyletin	C_14_H_12_O_3_	228	*P. anthriscoides* *P. anthriscoides*	[65] [65]
274	isopimpinellin	C_13_H_10_O_5_	246	*P. anisum*	[70]
275	methoxsalen	C_12_H_8_O_4_	216	*P. anisum*	[70]
276	umbelliprenin	C_24_H_30_O_3_	366	*P. anisum*	[71]
277	**7-. prenyloxycoumarin**	C_14_H_14_O_3_	230	*P. anisum*	[72]
278	auraptene	C_19_H_22_O_3_	298	*P. anisum*	[72]
279	umbelliferone	C_9_H_6_O_3_	162	*P. anisum*	[72]
280	pimpinellin	C_13_H_10_O_5_	246	*P. anisum*	[73]

**Table 6 molecules-28-01571-t006:** Sterols of *Pimpinella* plants.

No.	Name	Formula	Mol. Wt.	Species	Reference
281	campesterol	C_28_H_48_O	400	*P. anisum*	[74]
282	*α*-spinasterol	C_29_H_48_O	412	*P. anisum*	[74]
283	stigmasta-5,7,22-trien-3-ol	C_29_H_46_O	410	*P. anisum*	[74]
284	Δ7-avenasterol	C_29_H_48_O	412	*P. anisum*	[74]
285	Δ5-avenasterol	C_29_H_48_O	412	*P. anisum*	[74]
286	Δ7-stigmastenol	C_29_H_50_O	414	*P. anisum*	[55]
287	Δ5,23-stigmastadienol	C_29_H_48_O	412	*P. anisum*	[55]
288	Δ7-campesterol	C_28_H_48_O	400	*P. anisum*	[77]
289	sitostanol	C_29_H_52_O	416	*P. anisum*	[77]
290	cycloartenol	C_30_H_50_O	426	*P. anisum*	[78]
291	24-methylenecycloartenol	C_31_H_52_O	440	*P. anisum*	[78]
292	*b*-sitosterol	C_30_H_52_O	428	*P. thellungiana* *P. candolleana* *P. brachycarpa*	[41] [75] [76]
293	*g*-sitosterol	C_29_H_50_O	414	*P. thellungiana*	[41]
294	stigmasta-5, 22-dien-3-olacetate	C_31_H_50_O_2_	454	*P. candolleana*	[49]
295	daucosterol	C_35_H_60_O_6_	576	*P. candolleana*	[49]
296	stigmasterol	C_29_H_48_O	412	*P. candolleana*	[75]
297	24*ζ*-methyl-5R-lanosta-25-one	C_30_H_52_O	428	*P. brachycarpa*	[76]
298	pregnenolone	C_21_H_32_O_2_	316	*P. brachycarpa*	[76]

**Table 7 molecules-28-01571-t007:** Organic acids of *Pimpinella* plants.

No.	Name	Formula	Mol. Wt.	Species	Reference
299	oleic acid	C_18_H_34_O_2_	282	*P. thellungiana*	[42]
300	palmitic acid	C_16_H_32_O_2_	256	*P. thellungiana* *P. aurea*	[42] [36]
301	2-methylbutanoic acid	C_5_H_10_O_2_	102	*P. thellungiana*	[34]
302	shikimic acid	C_7_H_10_O_5_	174	*P. thellungiana*	[79]
303	3,4-dihydroxybenzoic acid	C_7_H_6_O_4_	154	*P. thellungiana* *P. aurea*	[34] [36]
304	gallic acid	C_7_H_6_O_5_	170	*P. thellungiana* *P. aurea*	[80] [36]
305	3-*O*-*trans*-caffeoylquinic acid	C_16_H_18_O_9_	354	*P. thellungiana*	[63]
306	5-*O*-*trans*-caffeoylquinic acid	C_16_H_18_O_9_	354	*P. thellungiana* *P. brachycarpa*	[63] [82]
307	4-*O*-*trans*-caffeoylquinic acid	C_16_H_18_O_9_	354	*P. thellungiana* *P. brachycarpa*	[63] [82]
308	3,5-*O*-*trans*-dicaffeoylquinic acid	C_25_H_24_O_12_	516	*P. thellungiana* *P. brachycarpa*	[63] [82]
309	3,4-*O*-*trans*-dicaffeoylquinic acid	C_25_H_24_O_12_	516	*P. thellungiana* *P. brachycarpa*	[63] [82]
310	4,5-*O*-*trans*-dicaffeoylquinic acid	C_25_H_24_O_12_	516	*P. thellungiana* *P. brachycarpa*	[63] [82]
311	4-*O*-feruloylquinic acid	C_17_H_20_O_9_	368	*P. thellungiana*	[81]
312	1-*O*-feruloylquinic acid	C_17_H_20_O_9_	368	*P. thellungiana*	[81]
313	5-*O*-*trans*-coumaroylquinic acid	C_16_H_18_O_8_	338	*P. thellungiana* *P. brachycarpa*	[81] [82]
314	3-*O*-*trans*-caffeoyl-5-feruloylquinic acid	C_26_H_26_O_12_	530	*P. thellungiana*	[81]
315	4-*O*-*trans*-caffeoyl-5-feruloylquinic acid	C_26_H_26_O_12_	530	*P. thellungiana*	[81]
316	1-*O*-*trans*-caffeoyl-5-*O*-*trans*-coumaroylquinicacid.	C_25_H_24_O_11_	500	*P. brachycarpa*	[82]
317	1-*O*-*trans*-caffeoyl-5-*O*-7, 8-dihydro-7*α*-methoxycaffeoylquinic acid	C_26_H_28_O_13_	548	*P. brachycarpa*	[82]
318	1-*O*-7, 8-dihydro-7α-methoxycaffeoyl-5-*O*-*trans-*caffeoylquinic acid	C_26_H_28_O_13_	548	*P. brachycarpa*	[82]
319	1-*O*-*trans*-coumaroyl-5-*O*-*cis*-coumaroylquinic acid	C_25_H_24_O_10_	484	*P. brachycarpa*	[82]
320	1,5-di-*O*-*cis*-coumaroylquinic acid	C_25_H_24_O_10_	484	*P. brachycarpa*	[82]
321	1,5-*O*-*trans*-dicaffeoylquinic acid	C_25_H_24_O_12_	516	*P. brachycarpa*	[82]
322	5-*O*-*cis*-caffeoylquinic acid	C_16_H_18_O_9_	354	*P. brachycarpa*	[82]
323	4-*O*-*trans*-coumaroylquinic acid	C_16_H_18_O_8_	338	*P. brachycarpa*	[82]
324	5-*O*-*cis*-coumaroylquinic acid	C_16_H_18_O_8_	338	*P. brachycarpa*	[82]
325	5-hydroxyferulic acid	C_10_H_10_O_5_	210	*P. anisum*	[66]
326	ferulic acid	C_10_H_10_O_4_	196	*P. anisum*	[66]
327	sinapinic acid	C_11_H_12_O_5_	224	*P. anisum*	[66]
328	caffeic acid	C_9_H_8_O_4_	180	*P. anisum*	[66]
329	*p*-coumaric acid	C_9_H_8_O_3_	164	*P. anisum*	[66]
330	*trans-*cinnamic acid	C_9_H_8_O_2_	148	*P. anisum*	[66]
331	rosmarinic acid	C_18_H_16_O_8_	360	*P. anisum*	[66]
332	3-hydroxybenzoic acid	C_7_H_6_O_3_	138	*P. anisum*	[66]
333	salicylic acid	C_7_H_6_O_3_	138	*P. anisum*	[66]
334	4-hydroxybenzoic acid	C_7_H_6_O_3_	138	*P. anisum*	[66]
335	vanillic acid	C_8_H_8_O_4_	168	*P. anisum*	[66]
336	syringic acid	C_9_H_10_O_5_	198	*P. anisum*	[66]
337	3-phenyllactic acid	C_9_H_10_O_3_	166	*P. anthriscoides*	[65]
338	citric acid	C_6_H_8_O_7_	192	*P. anthriscoides*	[65]
339	tetradecanoic acid	C_14_H_28_O_2_	228	*P. diversifolia*	[53]
340	linoleic acid	C_18_H_32_O_2_	280	*P. diversifolia*	[53]
341	stearic acid	C_18_H_36_O_2_	284	*P. diversifolia*	[53]
342	dodecanoic acid	C_12_H_24_O_2_	200	*P. aurea*	[36]
343	pentadecanoic acid	C_15_H_30_O_2_	242	*P. aurea*	[36]

**Table 8 molecules-28-01571-t008:** The pharmaceutical effects of *Pimpinella* species.

Pharmaceutical Activity	Part	Extract/ Compound	Experimental Model	Species	Reference
**Antioxidant**	Seed	Essential oil (in vivo)	Favism rats	*P. anisum*	[7]
	Seed	Ethanol extract (in vivo)	GN induced nephrotoxicity in rats	*P. anisum*	[84]
	Seed	Essential oil	DPPH radical scavenging activity	*P. anisum*	[50] [85] [86]
	Seed	Nanostructuredessential oil	DPPH and ABTS scavenging activity	*P. anisum*	[87]
	Seed	Water extract	DPPH and ABTS scavenging activity; FRAP	*P. anisum*	[56]
	Seed	*N*-hexane extract	DPPH and ABTS scavenging activity; FRAP and β-carotene bleaching tests	*P. anisum*	[11]
	Seed	Polysaccharide	DPPH radical scavenging activity	*P. anisum*	[9]
	Seed	Fatty acids and phenolic compounds	DPPH and ABTS scavenging activity	*P. anisum*	[66] [78]
	Aerial part	Flavonoid glycosides (**230**–**237**)	DPPH radical scavenging activity; FRAP	*P. cappadocica*	[16]
	Aerial part	Flavonoid glycosides (**238**–**239**)	DPPH radical scavenging activity; FRAP	*P. rhodantha*	[17]
	Fruit	Essential oil	DPPH radical scavenging activity; *β*-carotene bleaching inhibition	*P. enguezekensis*	[51]
	Aerial part	Essential oil	DPPH and ABTS scavenging activity; phosphomolybdenum and FRAP	*P. anthriscoides*	[65]
	Aerial part	Ethyl acetate extract	DPPH scavenging activity (IC_50_ = 53.07µg/mL)	*P. alpina*	[88]
	Aerial part	Ethyl acetate extract	DPPH radical scavenging activity (IC_50_ = 74.9 µg/mL)	*P. affinis*	[89]
	Seed	3% essential oil	DPPH radical scavenging activity (IC_50_ =6.81 µg/mL), β-carotene bleaching inhibition (IC_50_ = 206 µg/mL), FRAP (EC_50_ =35.20 µg/mL)	*P. saxifraga*	[46] [90]
**Antibacterial**	Seed	Polysaccharide	*E. coli*, *P. aeruginosa*, *B. cerus*, and *S. aureus* (50 mg/mL)	*P. anisum*	[9]
	Fruit	Essential oil	*P. aeruginosa*	*P. anisum*	[90]
	Seed	Essential oil	*P. aeruginosa*	*P. anisum*	[8]
	Seed	Essential oil	*S. aureus* and *E. coli* biofilms	*P. anisum*	[91]
	Seed	Essential oil	*E. coli, P. aeruginosa, K. pneumonia, S. epidermidis, E. faecalis, S. pyogenes, B.cerus* and *S. aureus*	*P. anisum*	[92]
	Seed	Essential oil	*E amylovora* with MIC of 31.25 μg/ml	*P. anisum*	[93]
	Seed	Oil-based hydrogel	*C. albicans, C.glabrata* and *C. Parapsilosis.*	*P. anisum*	[94]
	Aerial part	Essential oil	*F.solani*, *S.brevicaulis, A spp*., *A. fumigatus* and *F. oxysporum* (MIC = 50–490 μg/mL)	*P. anisum*	[95]
	Seed	Combination oil with terbinafine	*T. rubrum* and *T. mentagrophytes*	*P. anisum*	[96]
	Seed	Essential oils	*T. rubrum*	*P. anisum*	[97]
	Seed	Essential oil	*A. niger, A. oryzae, M. pusillus and F. oxysporum*	*P. anisum*	[98]
	Seed	Essential oil	*C. perfringens* with MIC of 10 μg/ml	*P. anisum*	[99]
	Seed	Essential oil	*A. niger, A. oryzae,* and *A. ochraceus*	*P. anisum*	[100]
	Seed	Essential oil	*A. carbonarius*	*P. anisum*	[101]
	Seed	Oil-based PLA films	*L.* monocytogenes and *V.* parahaemolyticus	*P. anisum*	[102]
	Seed	Nanostructured oil	*Y. enterocolitica, B. cereus, E. coli, and L. monocytogenes*	*P. anisum*	[103] [104]
	Seed	Nanostructured oil	14 food infesting fungi: *A. sydowii, A. repens, A. fumigatus, A. niger, A. candidus, A. luchuensis, F. oxysporum, C. herbarum, F. poae, M. sterilia, C. lunata, A. humicola* and *A. alternata* at its MIC dose (0.08–0.5μL/mL)	*P. anisum*	[105]
	Seed	Nanostructured oil	*S. aureus, E. coli, C. albicans,* and *A. niger*	*P. anisum*	[87]
	Seed	Essential oil	*S. aureus* and *E. coli*	*P. alpine*	[88]
	Aerialpart	Essential oil	*B.cereus*, *S. typhimurium*, *M. luteus*	*P. saxifraga*	[46]
	Fruit	Essential oil	*A. lwoffii*, *E. coli*, *K. pneumonia*, *B. cereus*, *C. perfringens*, *S. pneumonia*, *C. krusei* and *C. albicans* (MIC: 5–75 mg/mL)	*P. enguezekensis*	[51]
	Aerial part	Essential oil	*S. parasitica* (MIC: 2 μg/mL, MFC: 4 μg/mL)	*P. affinis*	[52]
	Aerial part	Essential oil	*S. aureus, L. monocytogenes, B. cereus, S. typhimurium, P. aeruginosa, E. cloacae, E. coli, A. fumigatus, A. ochraceus, A. niger, A. versicolor, T. viride, P. funiculosum, P. ochrochloron, and P. verrucosum*	*P. anthriscoides*	[65]
**Anti-inflammatory activity**	Seed	Combination oil with terbinafine	LPS stimulated neutrophils	*P. anisum*	[96]
	Fruit	Anethole (**40**)	Croton oil-induced ear edema and carrageenan-induced pleurisy model	*P. anisum*	[106]
	Fruit	0.3% essential oil	LPS-treated HBEpC and HTEpC cells	*P. anisum*	[107]
	Fruit	Anethole (**40**)	PM_2.5_ induced BEAS-2B and HepG2	*P. anisum*	[108]
	Seed	BLAB tea	Oval bumin-induced allergic rhinitis model mice	*P. anisum*	[109]
	Seed	Polysaccharide	Model of foot swelling induced by carrageenan in mice	*P. anisum*	[9]
**Antitumor activity**	Seed	Essential oil	Cytotoxicity against Hep G2 cells	*P. anisum*	[110]
	Seed	Agnps containing aqueous extract	Cytotoxicity against human neonatal skin stromal cells and HT115 cells	*P. anisum*	[111]
	Seed	Agnps containing aqueous extract	Cytotoxicity against colorectal cancer cell lines	*P. anisum*	[112]
	Fruit	Pimpinelol (**205**)	Cytotoxicity against MCF-7 cell, IC_50_: 1.06 μM	*P. haussknechtii*	[10]
**Hypoglycemic activity**	Seed	Aqueous extract	Inhibitory activity against α-amylase and α-glucosidase, (IC_50_=692.6±5.2 and 73.9±2.2 μg/mL)	*P. anisum*	[11]
	Seed	Aqueous extract	Against pancreatic damage in STZ-induced diabetic rats	*P. anisum*	[113]
	Seed	Methanolic extract	Wound healing activity in STZ-induced diabetic rats	*P. anisum*	[114]
**Hypotensive activity**	Herb	Ethyl acetate and ethanol extract	Inhibition effect on angiotensin-converting enzyme (0.5–10mg/mL)	*P. brachycarpa*	[115]
	Seed	Aqueous extract	Calcium channel antagonist	*P. anisum*	[116]
**Insecticidal activity**	Seed	Essential oil	Insecticidal effect against *C. quinquefasciatus* (LC_50_ = 25.4 μL/L) and *S. littoralis* (LD_50_ = 57.3 μg/larva)	*P. anisum*	[117]
	Seed	Essential oil	Insecticidal effect against *L. dispar*	*P. anisum*	[118]
	Seed	Essential oil	Insecticidal effect against *P.truncatus* and *T. granarium*	*P. anisum*	[119]
	Seed	Essential oil	Insecticidal effect against *L. decemlineata*	*P. anisum*	[120]
	Seed	Essential oil	Insecticidal effect against *B. aeneus*	*P. anisum*	[121]
	Seed	Essential oil	Insecticidal effect against *T. castaneum* and *P. interpunctella*	*P. anisum*	[122]
	Seed	Essential oil	Insecticidal effect against *M. incognita*	*P. anisum*	[123]
	Seed	Essential oil	Insecticidal effect against *aphids*	*P. anisum*	[124]
	Seed	Essential oil	Insecticidal effect against *Ipstypographu*	*P. anisum*	[125]
	Seed	Essential oil	Insecticidal effect against *P. citri*	*P. anisum*	[126]
	Seed	Essential oil	Insecticidal effect against *A. obtectus*	*P. anisum*	[127]
	Seed	Essential oil	Acaricidal and reduce AchEin two-spotted spider mite	*P. anisum*	[86]
	Seed	Essential oil nano-emulsions	Insecticidal effect against *S. oryzae* and *T. castaneum*	*P. anisum*	[128]
	Seed	Essential oil nano-emulsions	Insecticidal effect against *B. oleae*	*P. anisum*	[129]
	Seed	Essential oil nano-emulsions	Insecticidal effect against *M. persicae*	*P. anisum*	[130]
	Seed	Essential oil nano-emulsions	Insecticidal effect against *T. confusum*	*P. anisum*	[131]
	Seed	Essential oil nano-emulsions	Insecticidal effect against *T. castaneum*	*P. anisum*	[132] [133]
	Seed	Essential oil	Insecticidal effect against *Aedes aegypti*	*P. anisum*	[134]
	Seed	Essential oil	Insecticidal effect against *Musca domestica*	*P. anisum*	[135]
	Seed	Essential oil	Insecticidal effect against *Ixodes ricinus*	*P. anisum*	[136]
	Seed	Essential oil	Insecticidal effect against *Trypanosoma brucei*	*P. anisum*	[46]
	Seed	Essential oil	Insecticidal effect against *Culexquinquefasciatus*	*P. anisum*	[137] [138]
	Seed	*P*-anisaldehyde (**47**)	Insecticidal effect against horn fly, *H.irritansirritans*	*P. anisum*	[139]
	Seed	*P*-anisaldehyde (**47**)	Insecticidal effect against lone star tick, *A. americanum*	*P. anisum*	[140]
	Seed	Essential oil-loaded zein nanocapsules	Insecticidal effect against mosquito	*P. anisum*	[141]
**Enzymes inhibitory activity**	Aerial part	95% ethanol extract	Inhibitory effects on CYP1A2, 2A6, 2B6, 2C9, 2C19, 2D6, 2E1, and 3A4 in human liver microsomes	*P. brachycarpa*	[142]
	Aerial part	Essential oil	Tyrosinase, *α*-amylase, *α*-glucosidase, AChE and BChE	*P. anthriscoides*	[65]
	Seed	Water extract	Inhibitory effects on GSTA1–1(IC_50_ = 3.40 ± 0.83μg/mL)	*P. anisum*	[143]
	Seed	Essential oil	Inhibitory effects on xanthine oxidase (IC_50_ = 2.37 ± 0.23 μg/mL)	*P. anisum*	[144]
	Seed	Ethanol extract	Selective modulators of RALDHs	*P. anisum*	[145]
	Root	Bergapten (**256**) isopimpinellin (**274**) methoxsalen (**275**)	Inhibitory effects on CYP 1A2	*P. anisum*	[70]
	Seed	Phenolic extract	Inhibitory effects on AChE and BChE (IC_50_ = 0.07 and 0.34 μg/mL)	*P. anisum*	[56] [66]
**Anti depressant activity**	Seed	70% ethanol total extract (100 mg/kg)	Antidepressant and anxiolytic effects on Swiss Albino mice	*P. anisum*	[145]
	Seed	Essential oil	Memory impairment, anxiety, and depression in scopolamine-induced rats	*P. peregrina*	[18]
	Seed	Essential oil	Inhibition of brain cerebral cortex and hippocampus inflammation	*P. anisum*	[146]
	Seed	Essential oil	Inhibition of brain cerebral cortex and hippocampus antioxidant effects	*P. anisum*	[147]
	Herb	Extract	Clinical treatment of depression in patients with IBS	*P. anisum*	[148]
**Uterine relaxant activity**	Seed	50%hidroalcoholic extract	Uterine contraction induced by oxytocin, Bay K8644, carbachol, or generated spontaneously	*P. anisum*	[140]
**Wound healing effect**	Seed	Polysaccharide	Reparation of laser burn wounds in mice	*P. anisum*	[9]
**Migraine headache**	Herb	Essential oil	Clinical treatment of migraine	*P. anisum*	[141]
**Premenstrual syndrome**	Herb	Extract	Clinical treatment of premenstrual syndrome	*P. anisum*	[142]
**Skin whitening effect**	Herb	Umbelliprenin (**276**)	Melan-a cells of mice	*P. anisum*	[71]

## Data Availability

Not applicable.

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
