# Peer review of "Emerging Biopharmaceuticals from Pimpinella Genus"

_molecules, 2023, doi:10.3390/molecules28041571_

Round 1

Reviewer 1 Report

1.      In the abstract, authors should mention how many species of Pimpinella they researched on, and which species are most common, bioactive….etc.

2.      Keywords, all of them are so general, authors should specify some keywords.

3.      Authors are using uncommon words make it not clear, i.e.,  triumpantic, ceded, fringe, armenmantrum, dampned, and so on, English editing is required

4.      Authors should mention the novelty of their review as similar review has been recently published.

Nasır, Abir, and Erdal Yabalak. "Investigation of antioxidant, antibacterial, antiviral, chemical composition, and traditional medicinal properties of the extracts and essential oils of the Pimpinella species from a broad perspective: a review." Journal of Essential Oil Research 33, no. 5 (2021): 411-426.

5.      Reference 12 cannot find, please add the link if on published paper.

6.      Authors should revise the instructions of author, and unify the font size, spaces, underlines …..etc

7.      In the abstract the author mention "In this review, we attempt to summarize the present knowledge on the traditional applications, phytochemistry and pharmacology of genus Pimpinella from 2015 to 2022.  " But in the conclusion, this review is prepared to provide an overview to the knowledge of last 14 years (since 2000) !!!

Author Response

Reviewer 1

Dear Reviewer, thank you so much for your in-depth comments and give us a chance to polish our manuscript. We strongly believe that critical comments from reviewers always enhance the quality of the manuscript and make it more attractive for the readers. We have carefully revised the manuscript and responded to your comments point by point and also in the manuscript. All the changes in the manuscript have been highlighted through track changes.

  1. In the abstract, authors should mention how many species of Pimpinella they researched, and which species are most common, bioactive….etc.

Response to 1: Dear Reviewer, as per your suggestion we have added the species information and their pharmacological potential in the abstract.

  1. Keywords, all of them are so general, authors should specify some keywords.

Response to 2: Dear reviewer, we apologize for this negligence and we have changed the keywords, more matching our article.

  1. Authors are using uncommon words make it not clear, i.e.,triumpantic, ceded, fringe, armenmantrum, dampned, and so on, English editing is required.

Response to 3: Dear Reviewer, thank you so much for your suggestion, we have carefully revised the article from top to bottom line by line and removed all types of English mistakes including grammatical, typographical, and language errors, and used an easy synonym for all the above words. The word armenmantrum was a typo mistake its actual word is “armamentarium” which is a specific word used for pharmaceuticals.

  1. Authors should mention the novelty of their review as similar review has beenrecently published.Nasır, Abir, and ErdalYabalak. "Investigation of antioxidant, antibacterial, antiviral, chemical composition, and traditional medicinal properties of the extracts and essential oils of the Pimpinella species from a broad perspective: a review." Journal of Essential Oil Research 33, no. 5 (2021): 411-426.

Response to 4: Dear reviewer, once again we are thankful to you for investigating our studies in such in-depth and cross-studding our article with other related articles. We have carefully read Abir Nasır’s review and we hereby suggest some points regardingthe novelty of our review.

In Abir Nasır’s review, the authors have only focused on the chemical constituents and pharmacological activities of volatile oil extracted from different parts of Pimpinella plants. But in our article, we mentioned all the chemical constituents and pharmacological activities of various types of extracts including (aqueous, methanol, ethyl acetate, volatile oil, etc.) from different parts of Pimpinella plants. The chemical constituents in our article are more comprehensive and the structural types are more diverse. Furthermore, in Abir Nasir’s article, the authors only discussed the common pharmacological activities of Pimpinella plants, including antioxidant, antibacterial, cytotoxic, antiviral, and anti-COVID-19 activities, While our review systemically discussed some novel pharmacological activities, such as anti-tumor, anti-depression, hypotensive, hypoglycemic and liver protection, which have gradually attracted attention and investigation recently, providing a theoretical basis for further development of Pimpinella species.

  1. Reference 12 cannot find, please add the link if on published paper.

Response to 5: Dear Reviewer, Sorry for the mistake, Actually the references are prepared through References software (Mendeley) and sometimes this software automatically abrupt some bibliography data. But now we have provided the link to this published paper and also re-evaluated the whole bibliography information.

  1. Authors should revise the instructions of author, and unify the font size, spaces, underlines …..etc

Response to 6: Dear Reviewer, Sorry once again for our typo mistakes, we have carefully revised the manuscript as per your suggestion and as per journal instructions.

  1. In the abstract the author mention "In this review, we attempt to summarize the present knowledge on the traditional applications, phytochemistry and pharmacology of genus Pimpinella from 2015 to 2022." But in the conclusion, this review is prepared to provide an overview to the knowledge of last 14 years (since 2000) !!!

Response to 7: Dear Reviewer, it was a typo mistake. We have carefully revised it.

Reviewer 2 Report

The review entitled "Emerging biopharmaceuticals from the Pimpinella genus " refers to the pharmacological applications of the Pimpinella genus. Moreover, this review considers each aspect of the topic in detail and gives a general account of the latest works considered (2015–2022). 

However, there are some issues:

• In the introduction, in the caption of table 1 and in some parts of the text, words are underlined and written larger. All the text should be written in the same script and of the same size, without underlining. • In line 95, the sentence refers to gastrointestinal dysfunction, asthma and cough; the cited articles do not mention the same topics. • The text has many typos: line 175 “Spain to” or line 454 “against”. • Sort table 1 by part, please. • Please insert the headings of table 3, 6, 7, and 8 for each page including the table. • In the text and in Figure 9, molecular mechanisms are mentioned but  not explained or subsequently reported. The mechanisms of action involving IL-1 beta, IL-6, TNF-alpha, etc. should be better explained. • The paragraph “Hypoglycaemic activity” is unclear, how does P. anisum affect diabetes? How was the immune response improved? Moreover, “b-cell” line 468 should be write in capital letter.  • The name of the author cited in the text should be depicted with the surname, initials of the name and the words et al. • Table 8 should be better organised • Generally, the entire article needs a deep linguistic revision

Author Response

The review entitled "Emerging biopharmaceuticals from the Pimpinella genus " refers to the pharmacological applications of the Pimpinella genus. Moreover, this review considers each aspect of the topic in detail and gives a general account of the latest works considered (2015–2022). 

 Dear Reviewer, thank you so much for your in-depth comments and give us a chance to polish our manuscript. We strongly believe that critical comments from reviewers always enhance the quality of the manuscript and make it more attractive for the readers. We have carefully revised the manuscript and responded to your comments point by point and also in the manuscript. All the changes in the manuscript have been highlighted through track changes

However, there are some issues:

  1. In the introduction, in the caption of table 1 and in some parts of the text, words are underlined and written larger. All the text should be written in the same script and of the same size, without underlining

Response to 1: Dear reviewer, we apologize for our typo mistakes. We have carefully revised the whole manuscript from start to end line by line for all kinds of mistakes including (Typo, grammatical and language errors).

  1. In line 95, the sentence refers to gastrointestinal dysfunction, asthma and cough; the cited articles do not mention the same topics.

Response to 2: Accepted and thanks. In reference 13, the author mentions, "In particular, this plant has been used in Korean folk medicine for treating gastrointestinal disturbances, bronchial asthma, insomnia, and persistent cough." It is proved that Pimpinella brachycarpa can be used locally in Korea to treat the above diseases.

  1. The text has many typos: line 175 “Spain to” or line 454 “against”.

Response to 3: Accepted and thanks. We have corrected the spelling of “Spainto to” to “Spain to”, and corrected the spelling of “gainst” to “aginst”. We have carefully revised the whole manuscript from start till end line by line for all kinds of mistakes including (Typo, grammatical and language errors).

  1. Sort table 1 by part, please.

Response to 4: Dear reviewer, as per your suggestion, we have revised table 1 and reclassified it by parts.

  1. Please insert the headings of table 3, 6, 7, and 8 for each page includingthe table.

Response to 5: Accepted and thanks. We carefully checked the tables in full text and inserted headings on each page for the cross-page tables.

  1. In the text and in Figure 9, molecular mechanisms are mentioned but not explained or subsequently reported. The mechanisms of action involving IL-1 beta, IL-6, TNF-alpha, etc. should be better explained.

Response to 6: Dear Reviewer as per your suggestion we have added a discussion on the action mechanisms involving IL-1b, IL-6, TNF-a in the parts of "Antidepressant activity", which could give a better explanation of the antidepressant mechanism of P. anisum volatile oil.

  1. The paragraph “Hypoglycaemicactivity”isunclear, how doesP.anisumaffect diabetes?How was the immune response improved?Moreover, “b-cell” line 468 should be writed in capital letter.

Response to 7: Dear Reviewer as per your suggestion we have revised the discussion on part of “Hypoglycaemic activity”. And this study observed that b-cell structure was significantly improved, insulin immune response was enhanced, and pancreatic acinus and amylase levels were reduced in the P. anisum-treated group compared to diabetic-control. The authors attributed the beneficial effects of P. anisum extract to its hypoglycemic and antioxidant properties, as oxidative stress plays a critical role in the development and progression of diabetes. In this study, the P. anisum-treated group significantly reduced SOD and CAT and increased the level of lipid peroxidation marker MDA, which plays a role in lowering blood glucose. In addition, in immunohistochemical experiments, it could be observed that compared with diabetic control groups, the caspase3 immunoreaction (22.34±1.27 vs. 52.96±2.32) and beclin 1 immunoreaction (31.55±1.05 vs. 46.85±1.30) were significantly decreased in the P. anisum-treat group (p < 0.001). These results indicated that P. anisum could significantly down-regulate the autophagy regulation marker beclin 1 and apoptosis marker caspase 3 in pancreas, also relating to its antioxidant properties.

  1. The name of the author cited in the text should be depicted with the surname, initials of the name and the words et al.

Response to 8: Dear Reviewer as per your suggestion we have checked the whole text carefully and corrected all names of the author cited in the text.

  1. Table 8 should be better organized.

Response to 9: Dear Reviewer as per your suggestion we have organized Table 8 in a better way and also removed some typos and grammatical mistakes.

Round 2

Reviewer 1 Report

Yes, it has been modified according to our suggestions.